# Polysaccharide breakdown products drive degradation-dispersal cycles of foraging bacteria through changes in metabolism and motility

Astrid Katharina Maria Stubbusch[1,2,3]*, Johannes M Keegstra[4], Julia Schwartzman[5,6], Sammy Pontrelli[7], Estelle E Clerc[4], Samuel Charlton[4], Roman Stocker[4], Cara Magnabosco[3], Olga T Schubert[1,2], Martin Ackermann[1,2,8], Glen G D'Souza[1,2]*

[1]Institute of Biogeochemistry and Pollutant Dynamics, Department of Environmental Systems Science, ETH Zurich, Zurich, Switzerland; [2]Department of Environmental Microbiology, Eawag: Swiss Federal Institute of Aquatic Science and Technology, Dübendorf, Switzerland; [3]Geological Institute, Department of Earth Sciences, ETH Zurich, Zurich, Switzerland; [4]Institute of Environmental Engineering, Department of Civil, Environmental and Geomatic Engineering, ETH Zurich, Zurich, Switzerland; [5]Department of Civil and Environmental Engineering, MIT, Cambridge, United States; [6]Department of Biology, University of Southern California, Los Angeles, United States; [7]Institute of Molecular Systems Biology, Department of Biology, ETH Zurich, Zurich, Switzerland; [8]Laboratory of Microbial Systems Ecology, School of Architecture, Civil and Environmental Engineering (ENAC), École Polytechnique Fédéral de Lausanne (EPFL), Lausanne, Switzerland

*For correspondence:
astubbusch@icloud.com (AKMS);
glengeralddsouza@gmail.com
(GGD'S)

Competing interest: The authors declare that no competing interests exist.

## eLife assessment

This manuscript is a **valuable** contribution to our understanding of foraging behaviors in marine bacteria. The authors present a conceptual model for how a marine bacterial species consumes an abundant polysaccharide. Using experiments in microfluidic devices and through measurements of motility and gene expression, the authors offer **convincing** evidence that the degradation products of polysaccharide digestion can stimulate motility.

**Abstract** Most of Earth's biomass is composed of polysaccharides. During biomass decomposition, polysaccharides are degraded by heterotrophic bacteria as a nutrient and energy source and are thereby partly remineralized into $CO_2$. As polysaccharides are heterogeneously distributed in nature, following the colonization and degradation of a polysaccharide hotspot the cells need to reach new polysaccharide hotspots. Even though many studies indicate that these degradation-dispersal cycles contribute to the carbon flow in marine systems, we know little about how cells alternate between polysaccharide degradation and motility, and which environmental factors trigger this behavioral switch. Here, we studied the growth of the marine bacterium *Vibrio cyclitrophicus* ZF270 on the abundant marine polysaccharide alginate, both in its soluble polymeric form as well as on its breakdown products. We used microfluidics coupled to time-lapse microscopy to analyze motility and growth of individual cells, and RNA sequencing to study associated changes in gene expression. We found that single cells grow at reduced rate on alginate until they form large groups that cooperatively break down the polymer. Exposing cell groups to digested alginate accelerates

cell growth and changes the expression of genes involved in alginate degradation and catabolism, central metabolism, ribosomal biosynthesis, and transport. However, exposure to digested alginate also triggers cells to become motile and disperse from cell groups, proportionally increasing with the group size before the nutrient switch, and this is accompanied by high expression of genes involved in flagellar assembly, chemotaxis, and quorum sensing. The motile cells chemotax toward polymeric but not digested alginate, likely enabling them to find new polysaccharide hotspots. Overall, our findings reveal cellular mechanisms that might also underlie bacterial degradation-dispersal cycles, which influence the remineralization of biomass in marine environments.

## Introduction

Polysaccharides represent the largest fraction of biomass on Earth (*BeMiller, 2019*; *Reintjes et al., 2019*) and are constantly degraded and remineralized by microorganisms. Polysaccharides, also known as glycans, are long chains of monosaccharide units produced by cells for structural support (e.g. cellulose, chitin, and alginate) or energy storage (e.g. starch or glycogen; *BeMiller, 2019*). Heterotrophic microbes obtain nutrients and energy from the polysaccharide breakdown products. They often use exoenzymes, either secreted or anchored in the cell membrane (*Reintjes et al., 2019*; *Ratzke and Gore, 2016*), to cleave these large polymers into smaller units that can be taken up by cells. The formation of dense cell groups is observed during growth on polysaccharides by diverse bacteria, including the soil- and gut-dwelling *Bacillus subtilis* (*Ratzke and Gore, 2016*), the oligo-trophic fresh-water *Caulobacter crescentus* (*Povolo et al., 2022*), and several representatives of the copiotrophic (*Westrich et al., 2018*; *Takemura et al., 2014*) marine *Vibrio spp.* (*Schwartzman et al., 2022*; *D'Souza et al., 2023a*). It has been suggested that 'cooperative' growth (*Ratzke and Gore, 2016*), where dense cell groups reduce diffusional loss of valuable degradation products and secreted exoenzymes, represents a general principle in polysaccharide degradation (*Povolo et al., 2022*; *D'Souza et al., 2023a*). Yet, how bacteria subsequently disperse from exhausted polysaccharide sources and navigate towards new polysaccharide hotspots remains poorly understood.

To study the cellular mechanisms that govern the switch between polysaccharide degradation and dispersal toward new polysaccharide hotspots, we worked with the ubiquitous marine Gammaproteobacterium *Vibrio cyclitrophicus* ZF270, a degrader of the prevalent marine polysaccharide alginate (*Takemura et al., 2014*; *Wang et al., 2020*). Alginate is a linear polysaccharide that is produced by brown algae as a cell wall component, as well as by certain bacteria. It can be cleaved into oligomers by endo-acting alginate lyases or into monomers by exo-acting alginate lyases (alginate lyases reviewed here *Wong et al., 2000*), namely β-D-mannuronic acid and α-L-guluronic acid (*Mabeau and Kloareg, 1987*). *V. cyclitrophicus* ZF270 is found predominantly in the large-particle fractions of coastal water (*Hunt et al., 2008*), can attach and form biofilms on the surfaces of multiple polysaccharides including alginate (*Yawata et al., 2014*), and was shown to secrete alginate lyases during alginate degradation (*D'Souza et al., 2023a*). In this study, we used microfluidics coupled to automated time-lapse imaging to quantify the growth dynamics, group formation, and motility of *V. cyclitrophicus* ZF270 at the single-cell level under constant supply of either polymeric alginate or alginate degradation product in the form of digested alginate, as well as upon a transition from alginate to digested alginate. Furthermore, we used RNA-sequencing to compare the gene expression of *V. cyclitrophicus* ZF270 grown on alginate and digested alginate. We found striking responses to the form of alginate in growth rate, group formation, motility and chemotaxis, as well as in the expression of corresponding genes. Overall, our work provides insights into the metabolic and cellular regulation that allows cells to forage in heterogeneous nutrient-scapes through degradation-dispersal cycles.

## Results

### Extracellular break down of alginate delays population growth

To probe the phenotypic and metabolic regulation of bacterial cells during the progressive degradation of a polysaccharide source, we developed an experimental system consisting of the marine bacterium *V. cyclitrophicus* ZF270 growing on alginate, a highly abundant polysaccharide in the ocean. To mimic the local nutrient environment during the colonization of a new polysaccharide source, we used 0.1% weight per volume (w/v) algae-derived alginate in its soluble form (in the following also simply

referred to as 'alginate'). Since commercially available alginate breakdown products of specific sizes are limited and expensive, with only one monomeric component available at a considerable cost, we simulate an advanced stage of polysaccharide degradation by supplying cells with digested alginate (0.1% (w/v)). This digested alginate is prepared by treating alginate with a readily available endo-acting alginate lyase (see Materials and methods). This mimics an environment where degradation products like monomers and oligomers become abundant through the action of extracellular alginate lyases. The commercially available alginate lyase has been shown to produce alginate oligomers of progressively smaller size over extended digestion periods (*Huang et al., 2013*). We used liquid chromatography-mass spectrometry (LC-MS) to analyze the composition of the digested alginate, which had been subject to 48 hr of digestion. We found that digested alginate contained more mono-saccharides than untreated alginate, which contained more alginate molecules of higher molecular weight (*Figure 1—figure supplement 1*).

Using this system, we first set out to investigate the growth dynamics of *V. cyclitrophicus* ZF270 in well-mixed batch cultures containing either alginate or digested alginate as a sole carbon source (*Figure 1—figure supplement 2*). Measurements of optical density showed that the onset of growth on alginate was delayed by about 7.5 hr compared to growth on digested alginate (*Figure 1—figure supplement 2*). This is consistent with previous observations that in well-mixed environments the lag time of bacteria growing on alginate can be reduced by the external supplementation of alginate lyases (*D'Souza et al., 2023a*). We also observed that the optical density at stationary phase was higher when cells were grown on alginate (*Figure 1—figure supplement 2*). However, colony counts did not show a significant difference in cell numbers (*Figure 1—figure supplement 3*), suggesting that the increased optical density may stem from aggregation of cells in the alginate medium, as observed for other *Vibrio* species (*Schwartzman et al., 2022*). Overall, these findings indicate that growth on polysaccharides such as alginate in well-mixed cultures is initially limited by the extracellular breakdown of the polysaccharide, although similar cell numbers were reached eventually.

## Large cell groups form on alginate but not on digested alginate

To better understand how cells react to the changing degree of depolymerization of a polysaccharide source during degradation, we investigated the growth of *V. cyclitrophicus* ZF270 at the level of single cells on alginate and digested alginate. For this purpose, we grew the cells in microfluidic growth chambers, as depicted in *Figure 1A* and described in detail by *Dal Co et al., 2020*. In brief, the microfluidic chips are made of an inert polymer (polydimethylsiloxane) bound to a glass coverslip. The PDMS layer contains flow channels through which the culture medium is pumped continuously. Each channel is connected to several growth chambers that are laterally positioned. The dimensions of these growth chambers (height: 0.85 μm, length: 60 μm, width: 90–120 μm) allow cells to freely move and grow as monolayers. The culture medium, containing either alginate or digested alginate in their soluble form, is constantly pumped through the flow channel and enters the growth chambers primarily through diffusion (*Dal Co et al., 2020*; *D'Souza et al., 2021*; *Povolo et al., 2022*; *D'Souza et al., 2023b*; *D'Souza et al., 2023a*). Therefore, the number of cells and their positioning within microfluidic chambers is determined by the cellular growth rate as well as by cell movement (*Povolo et al., 2022*). This setup combined with time-lapse microscopy allowed us to follow the development of cell communities over time. We found that over 24 hr dense groups of more than 1000 cells formed on alginate, filling the entire microfluidic chamber (*Figure 1B*). In contrast, cells supplied with digested alginate grew in smaller groups that never exceeded 100 cells per chamber (*Figure 1C and D*). Reconstructed cell lineages revealed that the large cell groups on alginate formed because cells often did not disperse after division and thereby formed dense cell groups originating from a single cell lineage (*Figure 1B*). The lower cell density on digested alginate could be caused by slower growth or by more cells leaving the chambers. As the maximum growth rate both in bulk and on single cell level is similar in alginate and digested alginate (*Figure 1—figure supplement 2*), it is more likely that the lower cell density in digested alginate is caused by cell dispersal. To test the role of increased viscosity of poly-meric alginate in causing the increased aggregation of cells, we measured the viscosity of 0.1% (w/v) alginate or digested alginate dissolved in TR media. For alginate, the viscosity was 1.03±0.01 mPa·s (mean and standard deviation of three technical replicates) whereas the viscosity of digested alginate in TR media was found to be 0.74±0.01 mPa·s. Both these values are relatively close to the viscosity of water at this temperature (0.89 mPa·s *Berstad et al., 1988*) and, while they may affect swimming

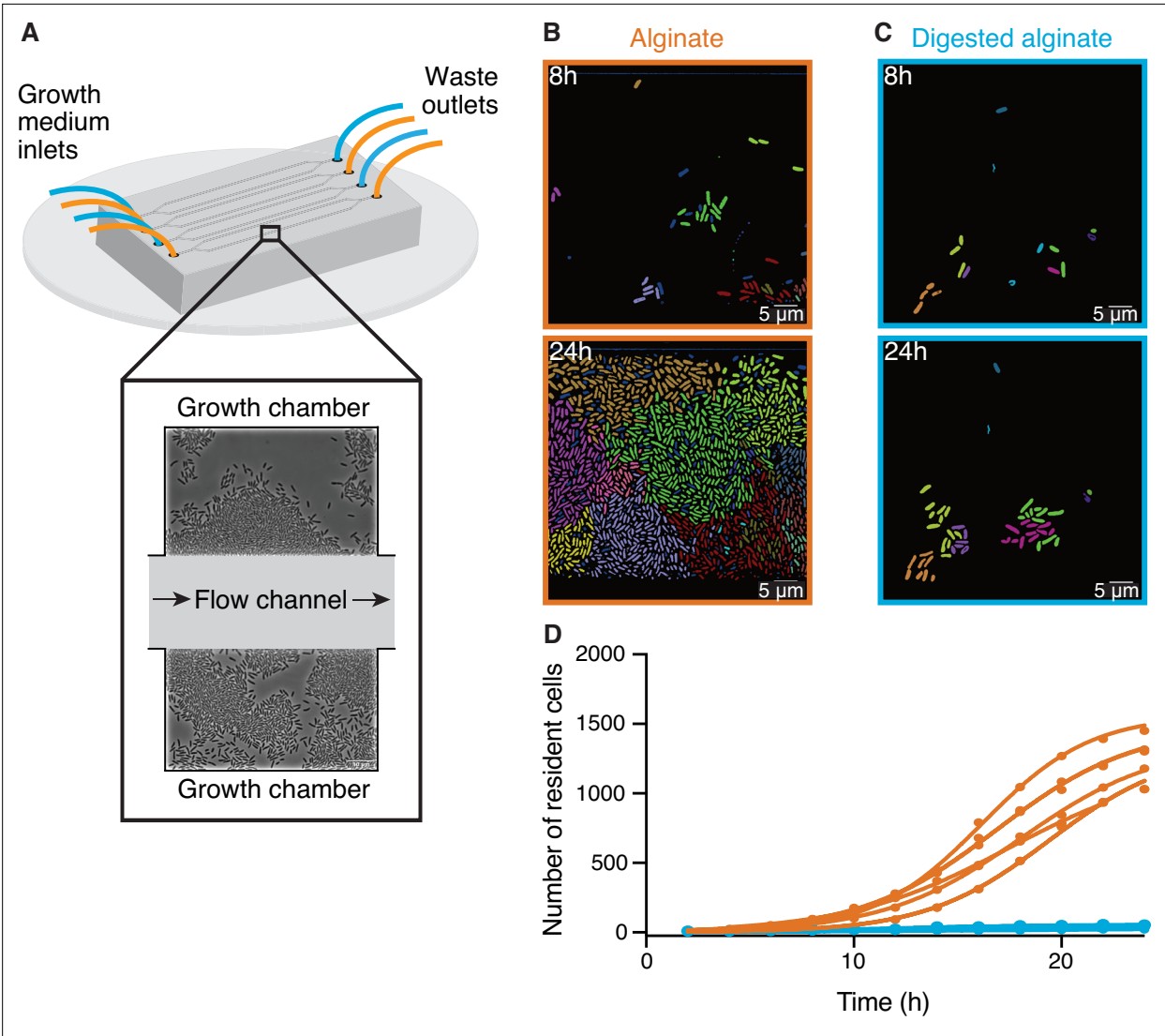

**Figure 1.** Large cell groups form on alginate but not on digested alginate. (**A**) Schematic representation of the setup of the microfluidic experiments. (**B and C**) Representative images at different time points of *V. cyclitrophicus* ZF270 cells growing in microfluidic chambers, described in detail by *Dal Co et al., 2020*, with (**B**) alginate medium or (**C**) digested alginate medium, both in their soluble form (not visible). Cells are false-colored according to their lineage identities based on cell segmentation and tracking over 24 hr. Cells without identified progenitors are colored in dark blue. See *Figure 1—video 1* (alginate) and *Figure 1—video 2* (digested alginate) for time-lapse videos. (**D**) Cell numbers within microfluidic chambers supplied with alginate (orange) are substantially higher than cell numbers within microfluidic chambers supplied with digested alginate (blue) (Logistic growth regression for alginate: $R^2$=0.99, maximal number of cells = 1217–1564, k=0.24–0.38 hr$^{-1}$; for digested alginate: $R^2$=0.86–0.97, maximal number of cells = –100, k=0.07–0.4 hr$^{-1}$). Circles indicate the number of cells present at a given time point in each chamber ($n_{chambers}$ = 7). Data for chambers with alginate originate from *D'Souza et al., 2023a*. Lines are fits of a logistic growth regression line for each condition.

The online version of this article includes the following video and figure supplement(s) for figure 1:

**Figure supplement 1.** Relative concentrations of the breakdown products of alginate after treatment with commercial alginate lyases.

**Figure supplement 2.** Polymeric alginate increases lag times and yield of *Vibrio cyclitrophicus* ZF270 populations.

**Figure supplement 3.** Cell counts of *V. cyclotrophicus* ZF270 on alginate and digested alginate measured by plating assay.

**Figure 1—video 1.** Time-lapse video of *Vibrio cyclitrophicus* ZF270 cells within a representative microfluidics chamber fed with 0.1% alginate as the sole carbon source.

https://elifesciences.org/articles/93855/figures#fig1video1

**Figure 1—video 2.** Time-lapse video of *Vibrio cyclitrophicus* ZF270 cells within a representative microfluidics chamber fed with 0.1% digested alginate as the sole carbon source.

https://elifesciences.org/articles/93855/figures#fig1video2

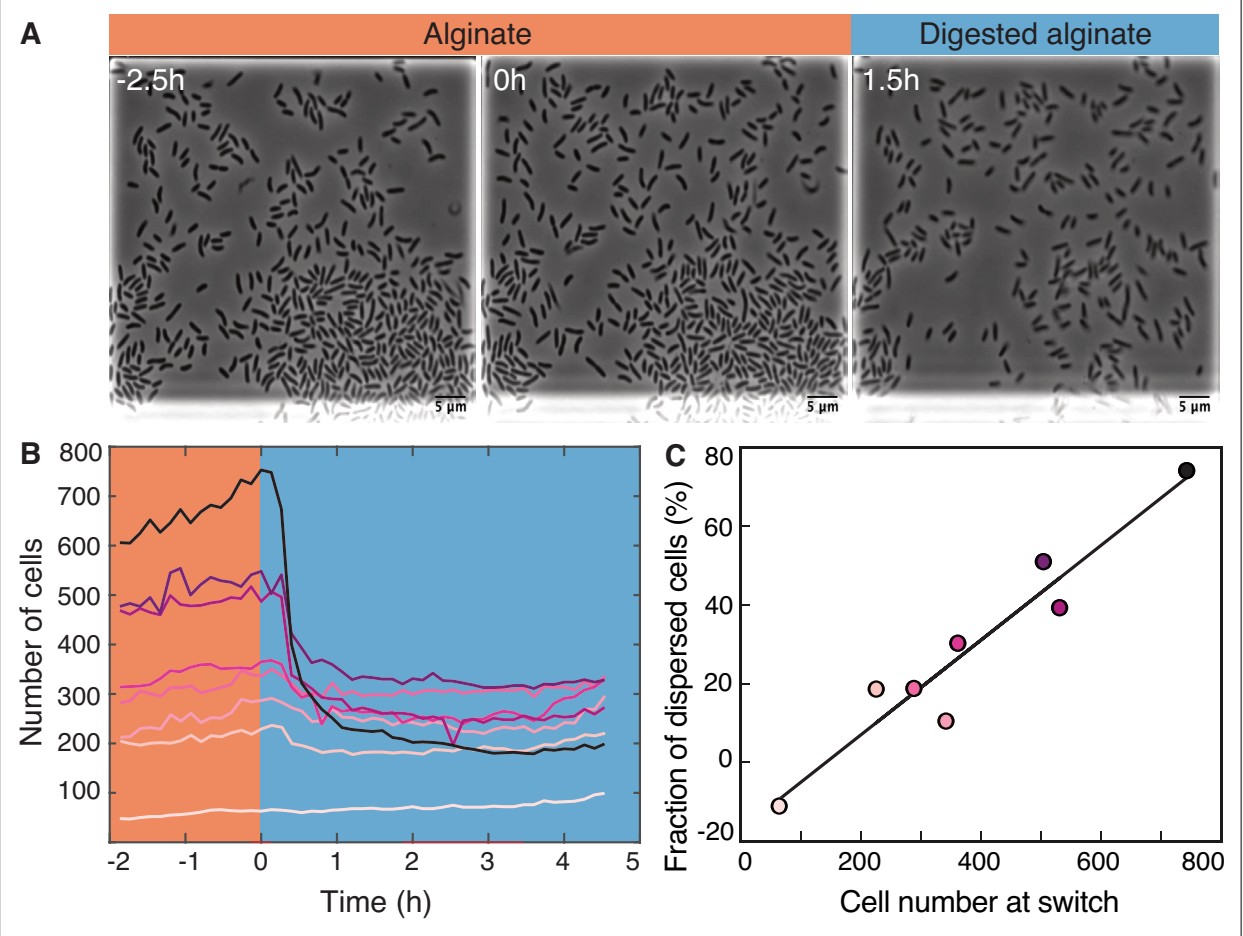

**Figure 2.** Transition from alginate to digested alginate triggers density-dependent dispersal of cells. (**A**) Representative time-lapse images of *V. cyclitrophicus* ZF270 cells (phase contrast microscopy) in microfluidic growth chambers that were initially exposed to alginate and then switched to digested alginate. (**B**) Number of cells in different chambers over time, each chamber indicated by a unique color (n=8). The carbon source is indicated by the colored background (orange: alginate; blue: digested alginate). See *Figure 2—video 1* for a time-lapse video. (**C**) Positive relationship between the number of cells in the microfluidic growth chamber at the time of the switch and the fraction of cells that disperse after the switch. Each circle represents one growth chamber with colors corresponding to (**B**), and the line depicts a linear regression fit ($R^2$=0.92, slope = 0.12).

The online version of this article includes the following video for figure 2:

**Figure 2—video 1.** Time-lapse video of *Vibrio cyclitrophicus* ZF270 cells within a representative microfluidics chamber fed with 0.1% alginate and then switched to 0.1% digested alginate as sole carbon sources.

https://elifesciences.org/articles/93855/figures#fig2video1

behavior (*Zöttl and Yeomans, 2019*), they are insufficient to physically restrain cell movement (*Berg and Turner, 1979*). Overall, our observations suggest that cells can modulate their propensity to form groups depending on the state of polysaccharide degradation in their local environment. Similar observations were made with a different model system (*Caulobacter crescentus* growing on the polysaccharide xylan; *D'Souza et al., 2021*), indicating that group formation on polymeric nutrient sources may be a general mechanism of bacteria that degrade polysaccharides extracellularly.

## Transition from alginate to digested alginate triggers density-dependent dispersal of cells

To investigate the transition from the large cell groups formed on alginate to the small groups formed on digested alginate, we subjected *V. cyclitrophicus* ZF270 cells grown on alginate to a switch to digested alginate (*Figure 2A*). Following the limited cell motility on alginate, this switch led to a rapid decrease in cell density within the growth chambers (*Figure 2B*), presumably caused by cell dispersal. As we previously reported for other *Vibrionaceae* isolates, the growth rate of the cells on

alginate was dependent on the local cell density: Initially, the growth rate increased with cell density but then decreased at high cell densities, indicating that cell groups can benefit from the sharing of breakdown products generated by each other's exoenzymes, but also increasingly compete for nutrients (*D'Souza et al., 2023a*). This led us to investigate whether cells in larger groups, potentially experiencing stronger nutrient competition, might have a higher propensity to disperse after a switch to digested alginate than cells in smaller groups. We indeed found that the nutrient switch caused a few or no cells to disperse from small cell groups (*Figure 2B*), whereas a large fraction of cells from large cell groups dispersed (*Figure 2C*). In fact, the fraction of cells that dispersed upon imposition of the nutrient switch showed a strong positive relationship with the number of cells present, meaning that cells in chambers with many cells were more likely to disperse than cells in chambers with fewer cells (*Figure 2C*). Thus, during the transition from polysaccharides to degradation products, we found that the dispersal rate of cells depends on the size of the cell groups, likely through increased motility of cells in large groups.

## Cells growing on digested alginate are more motile and polymeric alginate acts as chemoattractant

To investigate whether increased cell motility of *V. cyclitrophicus* ZF270 underlies the dispersal of cells observed after a shift to digested alginate, as also previously observed for *C. crescentus* cells on the monosaccharide xylose (*Povolo et al., 2022*; *D'Souza et al., 2021*), we quantified the motility of single cells supplied with either alginate or digested alginate. We found that the single-cell swimming speed as well as the swimming distance were significantly larger for cells supplied with digested alginate compared to cells supplied with alginate, and that a larger fraction of cells was motile (*Figure 3*). This confirmed increased motility as a cellular response to the exposure to degradation products in the form of digested alginate. Overall, these findings suggest that once the breakdown of a polysaccharide source makes breakdown products available in the local environment, a fraction of cells becomes motile and disperses.

To understand whether alginate polymers or alginate breakdown products act as chemoattractants on motile cells, we measured the chemotactic strength towards alginate and digested alginate using the In Situ Chemotaxis Assay (ISCA; *Lambert et al., 2017*; *Clerc et al., 2020*). Interestingly, we found *V. cyclitrophicus* ZF270 to significantly chemotax toward alginate (chemotactic index $I_c$ >1) but not significantly toward digested alginate (*Figure 3F*). This suggested that the increase in motility is accompanied by chemotaxis toward alginate polymers.

## Altered gene expression in central carbon metabolism, enzyme production, secretion and transporters, motility, and quorum sensing underlies the late-stage alginate degradation and cell dispersal

Next, we sought to elucidate the molecular mechanisms underlying the observed phenomenological disparities between cells cultivated on alginate and digested alginate. Due to the challenge of generating knock-out mutants in natural isolates, we used transcriptomics to investigate the differentially expressed genes of *V. cyclitrophicus* ZF270 under these respective conditions. To obtain a high-quality reference genome of *V. cyclitrophicus* ZF270, we sequenced and assembled a new reference genome using combined short and long read sequencing (BioProject PRJNA991487). We then grew cultures on either alginate or digested alginate until mid-exponential phase. To understand which cellular functions were affected by the expression changes, we first performed differential gene expression analysis using the software DESeq2. Here, genes exhibiting a log2 fold expression change greater than 0.5 or smaller than –0.5 between the two conditions, with a Benjamini-Hochberg(BH)-adjusted p-value below 0.01, were considered to be differentially expressed (*Supplementary file 1*). Next, we investigated which KEGG categories were enriched in either genes with increased or decreased expression via Gene Set Enrichment Analysis (GSEA; *Subramanian et al., 2005*; *Supplementary file 2*) and found nine categories significantly enriched in genes with increased gene expression on digested alginate, and three categories significantly enriched in genes with decreased gene expression (*Figure 4A*).

Cells growing on digested alginate showed expression changes in parts of the central metabolism and translation, compared to cells growing on alginate. Specifically, two pathways of the central metabolism were significantly enriched in genes with lower expression on digested alginate ('Valine, leucine and isoleucine biosynthesis' and 'Propanoate metabolism' with a negative normalized

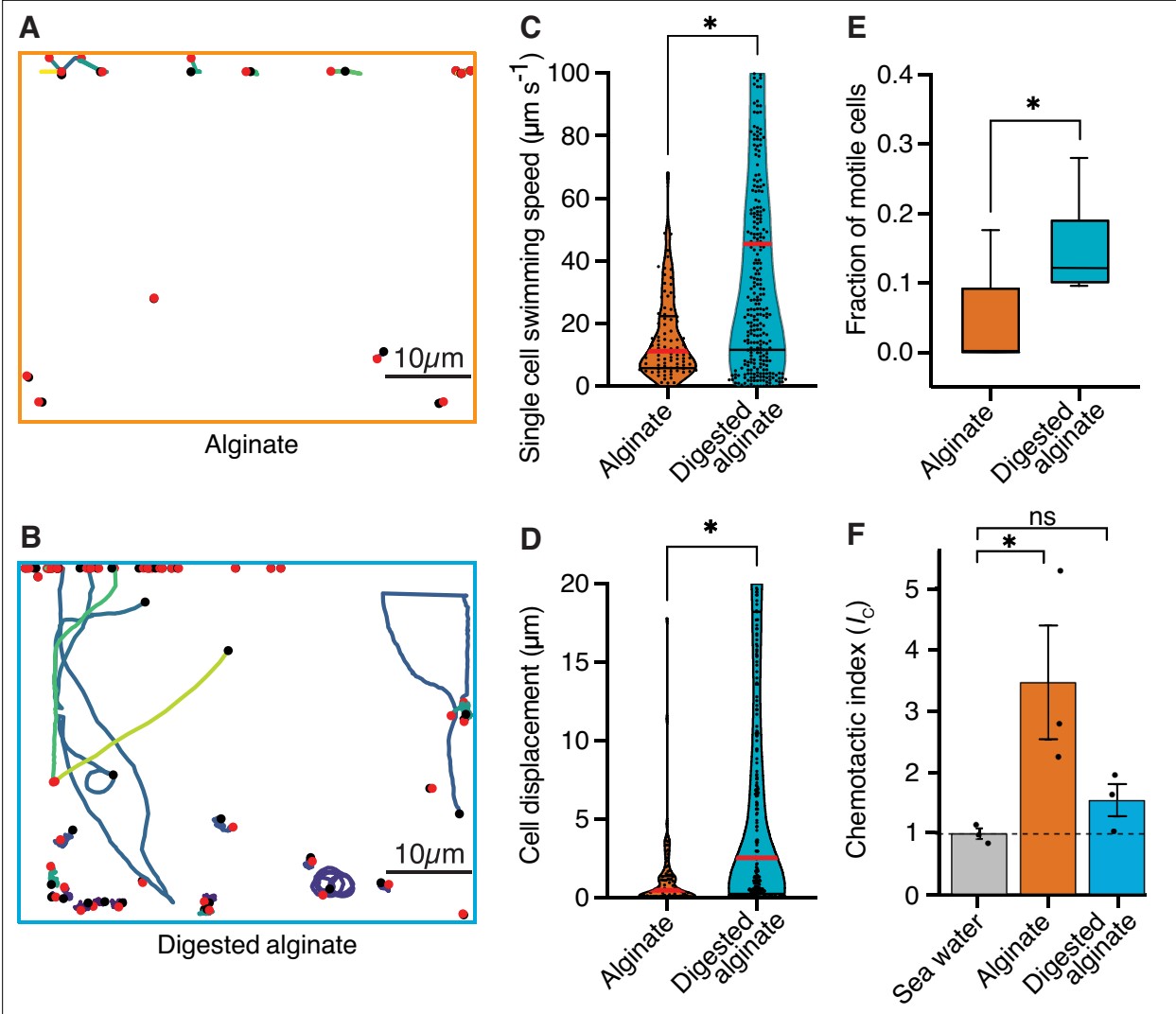

**Figure 3.** Cells are more motile on digested alginate than alginate and show chemotaxis towards alginate. (**A and B**) Spatial trajectories of cells supplied with (**A**) alginate or (**B**) digested alginate in representative microfluidic growth chambers are shown. Black points mark the starting point of each trajectory, pink points mark the end point of each trajectory, and colored lines mark the trajectories of cells. (**C**) Distributions of the mean single-cell swimming speeds (Nested t-test, p-value <0.0007, t=4.803, df = 10, $n_{cells}$ = 86 vs 375 in $n_{chambers}$ = 5) are shown. (**D**) Distributions of cell displacement over the course of a trajectory (Nested t-test test, p-value <0.0131, t=4.39, df = 10, $n_{cells}$ = 86 vs 375, and $n_{chambers}$ = 5) are shown. In (**C**) and (**D**) the red horizontal lines indicate the mean while black lines depict the 25th and 75th quartiles of the distribution. (**E**) The mean fraction of motile cells in each chamber, where motile cells are defined as cells with displacement greater than 1 μm (Mann-Whitney test on the means of five growth chambers, p-value = 0.034). In **C, D, and E**, each chamber was considered as an independent replicate. (**F**) Chemotactic index ($I_C$) quantified by In Situ Chemotaxis Assay (ISCA) (Tukey multiple comparisons of means, 95% family-wise confidence levels as error bars, p-value <0.05, n=3). Asterisks indicate statistically significant differences. See *Figure 3—video 1* and *Figure 3—video 1* for time-lapse videos of swimming cells.

The online version of this article includes the following video(s) for figure 3:

**Figure 3—video 1.** High frame rate (125 Hz, i.e. frames s⁻¹) time-lapse video of *Vibrio cyclitrophicus* ZF270 cells within a representative microfluidics chamber fed with 0.1% alginate as the sole carbon source.

https://elifesciences.org/articles/93855/figures#fig3video1

**Figure 3—video 2.** High frame rate (125 Hz, i.e., frames s⁻¹) time-lapse video of *Vibrio cyclitrophicus* ZF270 cells within a representative microfluidics chamber fed with 0.1% digested alginate as the sole carbon source.

https://elifesciences.org/articles/93855/figures#fig3video2

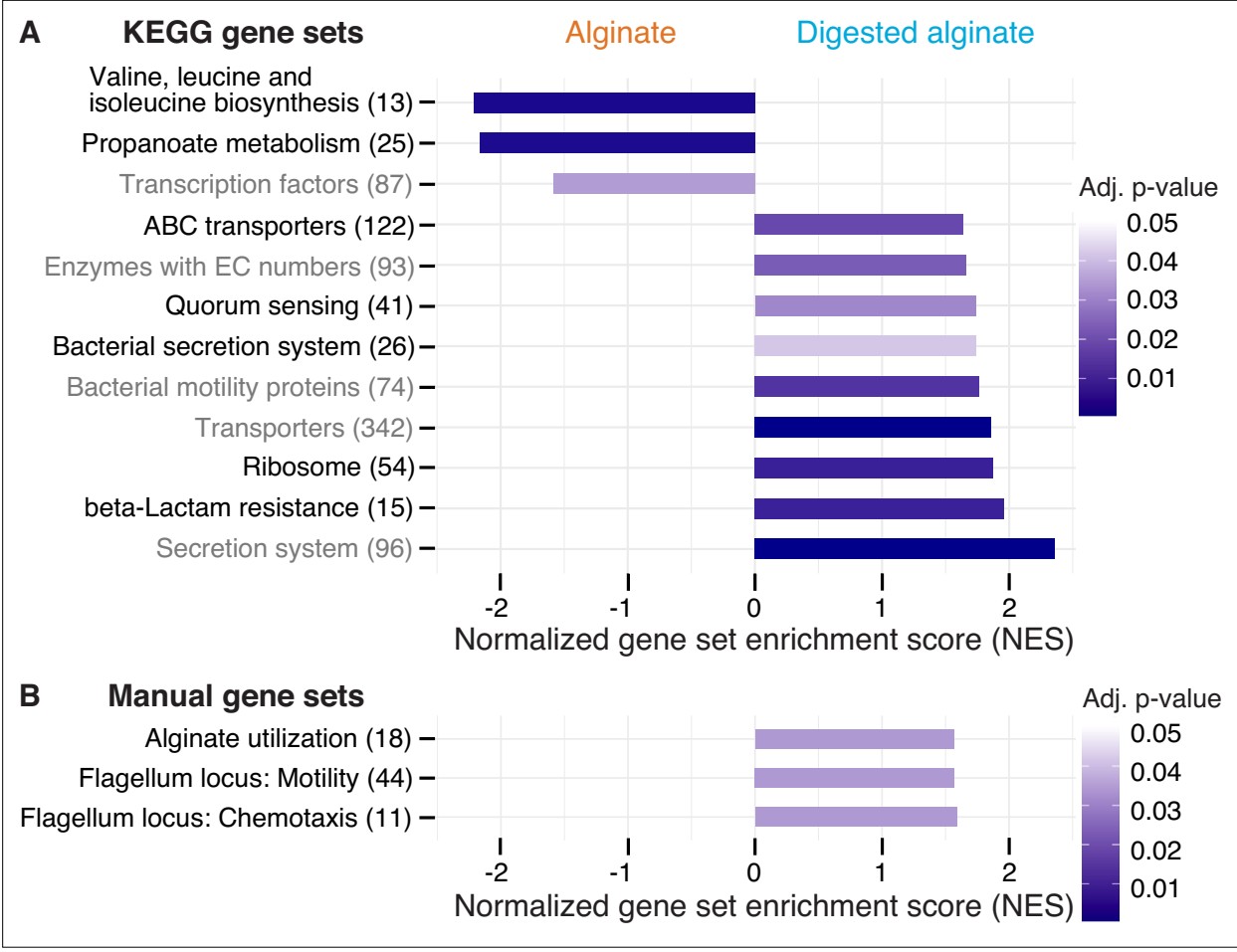

**Figure 4.** Twelve functional gene sets are enriched in genes with increased or decreased expression in cells grown on digested alginate. Gene set enrichment analysis (GSEA) with (**A**) all KEGG pathways and KEGG BRITE categories as gene sets or with (**B**) a custom alginate utilization, flagellar assembly, and flagellum-driven chemotaxis gene set was performed comparing the gene counts of the transcriptome of *V. cyclitrophicus* ZF270 cultures grown on digested alginate and alginate (six replicates each). Gene sets with a positive enrichment score were enriched with genes with higher expression in cells grown on digested alginate relative to cells grown on alginate (BH-adjusted p-value <0.05), whereas gene sets with negative enrichment scores were significantly enriched with genes with decreased expression on digested alginate. The number in brackets indicates the number of genes with unique K number per gene set (**A**) and the number of genes per gene set (**B**) within the *V. cyclitrophicus* ZF270 genome.

The online version of this article includes the following figure supplement(s) for figure 4:

**Figure supplement 1.** KEGG map of the significantly enriched KEGG pathway for valine, leucine and isoleucine biosynthesis.

**Figure supplement 2.** KEGG map of the significantly enriched KEGG pathway for propanoate metabolism.

**Figure supplement 3.** KEGG map of the significantly enriched KEGG pathway for ribosomal proteins.

**Figure supplement 4.** KEGG map of the significantly enriched KEGG pathway for bacterial secretion systems.

**Figure supplement 5.** KEGG map of the significantly enriched KEGG pathway for ABC transporters.

**Figure supplement 6.** KEGG map of the significantly enriched KEGG pathway for quorum sensing.

**Figure supplement 7.** KEGG map of the significantly enriched KEGG pathway for beta-lactam resistance.

enrichment score (NES), *Figure 4A*, *Figure 4—figure supplements 1 and 2*, *Supplementary file 3 and 4*). We also found the gene set that encodes ribosomal proteins significantly enriched with genes highly expressed on digested alginate (positive NES, *Figure 4A* and *Figure 4—figure supplement 3*, *Supplementary file 5*). This implies that cells invest proportionally more of their transcriptome into the production of new proteins, a sign of faster growth (*Wei et al., 2001*; *Gifford et al., 2013*). Both observations likely relate to the different growth dynamics of *V. cyclitrophicus* ZF270 on digested alginate compared to alginate (*Figure 1—figure supplement 3*), where cells in digested alginate medium reached their maximal growth rate 7.5 hr earlier and thus showed a shorter lag time (*Figure 1—figure*

*supplement 3*). As a consequence, the growth rate at the time of RNA extraction (mid-to-late exponential phase) may have differed, even though the maximum growth rate of cells grown in alginate medium and digested alginate medium were not found to be significantly different (*Figure 1—figure supplement 2*).

The expression of transporters and secretion systems was generally increased in cells growing on digested alginate, compared to cells growing on alginate. This includes genes with a function in 'Bacterial secretion systems', namely secretion systems I to VI, which mediate protein export through the inner and outer membranes of Gram-negative bacteria (*Figure 4A* and *Figure 4—figure supplement 4*, *Supplementary file 6*). Notably, 11 of the 13 General Secretion Pathway (GSP) genes (part of the type II secretion system) showed 1.1–3.4-fold increased expression levels on digested alginate (p-values <0.01, *Figure 4A* and *Supplementary file 8*). The GSP is known to facilitate secretion of various extracellular enzymes like chitinases, proteases, and lipases *Johnson et al., 2014*; *Sikora, 2013*, therefore, the positive enrichment on digested alginate may be linked to the export of extracellular alginate lyases. The KEGG BRITE category 'Secretion system' was also enriched on digested alginate (*Figure 4A*, *Supplementary file 7*). It contains additional genes involved in protein export, implicating them as interesting subjects to further research in their role in protein secretion and cell attachment and detachment during degradation-dispersal cycles of polysaccharide-degrading bacteria. Also, the gene set of transporters and in particular ABC transporters were positively enriched (*Figure 4A*, *Supplementary file 10 and 11*). The latter showed enrichment especially in genes related to saccharide, iron, zinc, and phosphate transport (*Figure 4—figure supplement 5*), which suggests that cells growing on digested alginate invest proportionally more of their transcriptome into uptake of not only saccharides but also essential nutrients like iron, zinc, and phosphate, which may become growth-limiting when degradation products are abundantly available.

Both motility and quorum sensing genes increased in expression in cells growing on digested alginate. The positive enrichment of the gene set containing bacterial motility proteins aligned with our expectations based on the increase in motile cells that we observed in *Figure 3E*; *Figure 4* and *Supplementary file 12*. The set of quorum sensing genes was also positively enriched in cells growing on digested alginate (*Figure 4A* and *Figure 4—figure supplement 6*, *Supplementary file 13*). This role in dispersal is in agreement with a previous study that showed induction of the quorum sensing master regulator in *V. cholerae* cells during dispersal from biofilms on a similar time scale as here (less than an hour; *Singh et al., 2017*). Quorum sensing is known to control biofilm formation in the well-studied model system *Vibrio cholerae* (*Jemielita et al., 2018*; *Waters et al., 2008*) and may also orchestrate the density-dependent dispersal in the presence of degradation products observed in our study, though the particular signaling cues remain to be uncovered. The strong cellular response to the degree of alginate depolymerization was also emphasized by the finding that transcription factor genes were enriched among genes with decreased expression on digested alginate (*Figure 4* and *Supplementary file 14*). The set of genes associated to KEGG's beta-lactam resistance category was enriched in genes with increased expression on digested alginate. However, most genes in this gene set are also associated to the KEGG pathways 'Transporters', 'Quorum sensing', 'Chromosome and associated genes', and 'Peptidoglycan biosynthesis' and thus the enrichment likely reflects expression changes in these categories and may relate to the difference in growth dynamics (*Figure 4A* and *Figure 4—figure supplement 7*, *Supplementary file 15*). Overall, the observed gene expression changes provide insights into the molecular mechanisms underlying the substantial adaptations of *V. cyclitrophicus* ZF270 cells to the polymeric and digested form of alginate.

## Growth on digested alginate is associated with increased expression of alginate catabolism, flagellar motility, and chemotaxis genes

For a more fine-grained understanding of the impact of the form of alginate on the alginate catabolism, we investigated specifically the expression of genes involved in alginate degradation, uptake, and catabolism. In the closed genome of *V. cyclitrophicus* ZF270, we identified genes that encode CAZymes responsible for alginate degradation, namely alginate lyase genes from the PL6, PL7, PL15, and PL17 family. We identified homologs of alginate transporters (porin *kdgM*, symporter *toaA*, *toaB*, and *toaC*) and metabolic enzymes that shunt into the Entner-Doudoroff pathway (DEHU reductase genes *dehR*, *kdgK*, and *eda*), based on the known genes in the alginate degradation pathway of *Vibrionaceae* (*Wargacki et al., 2012*; *Zhang et al., 2021*). We found that the expression of most of

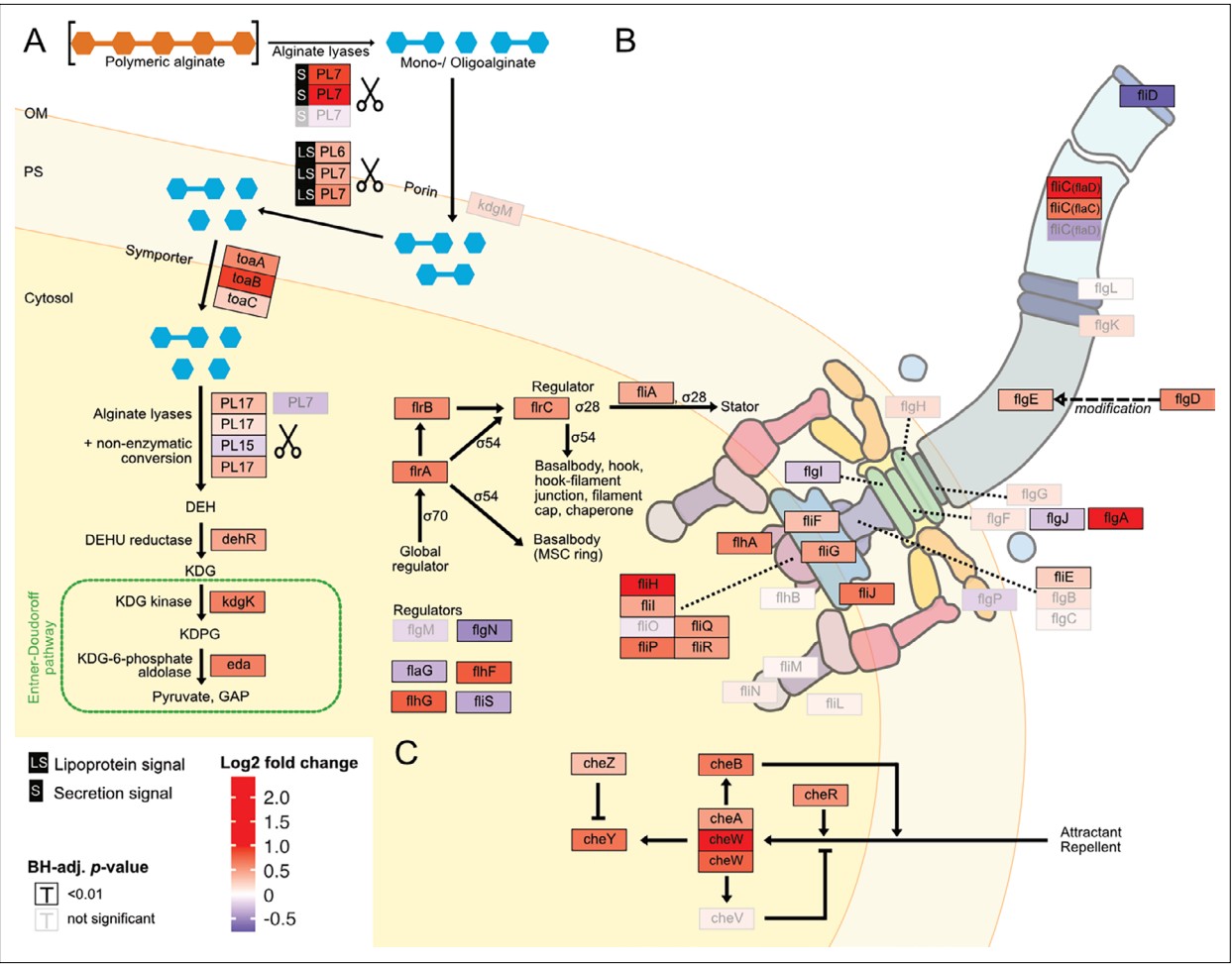

**Figure 5.** Digested alginate increases expression of genes involved in alginate degradation, uptake and catabolism, as well as flagellar assembly and chemotaxis. Genome-wide differential expression analysis where the log2 fold changes of gene expression on digested alginate compared to alginate is shown for (**A**) alginate lyases (*PL6, PL7, PL15, PL17*, scissors symbol), transporters (porin *kdgM*, symporter *toaB*, symporter *toaC*), and metabolic enzymes shunting into the Entner-Doudoroff pathway (DEHU reductase *DehR*, *kdgK*, *eda*), (**B**) genes of the flagellar locus associated with flagellar assembly and (**C**) adjacent chemotaxis genes. Genes displayed in (**B**) and (**C**) are part of the KEGG pathways 'Bacterial motility proteins' and 'Bacterial chemotaxis'. Differential expression analysis was performed to compute the Benjamini-Hochberg-adjusted Wald test p-value ('BH-adj. p-value', text color and box outline color) and log2 fold change (box fill color) for each gene (box). For better visibility, genes that exhibited a log2 fold gene expression change greater than 1 (i.e. doubling of expression) or less than –1 (i.e. halving of expression) are designated maximum intensity of red or blue, respectively. Genes with BH-adj. p-value smaller than 0.01 were considered significantly differentially expressed. In (**A**), the location of the gene products was based on Figure 1 of *Wargacki et al., 2012* with the exception of the alginate lyases (*PL6, PL7, PL15, PL17*) which were placed based on their signal peptides (S: extracellular, LS: membrane-embedded, none: cytosolic). In (**B**) and (**C**) the gene location and depiction were based on the KEGG pathway 'Flagellar assembly' (map02040), 'Bacterial chemotaxis' (map02030), and Figure 3 of *Rajagopala et al., 2007*. Genes without known cellular location were omitted here but displayed in the genomic architecture in *Figure 5—figure supplement 1*. Arrow: activation; dashed arrow: modification; 'flat' arrow: inhibition; OM: outer membrane; PM: periplasm; IM: inner membrane; PL: polysaccharide lyase family; *kdgM*: oligogalacturonate-specific outer membrane porin; *toaABC*: oligoalginate symporter; DEH: 4-deoxy-L-erythro-5-hexoseulose uronic acid; *dehR*: DEH reductase; KDG: 2-keto-3-deoxy-gluconate; *kdgK*: KDG kinase; KDPG: 2-keto-3-deoxy-6-phosphogluconate; *eda*: KDG-6-phosphate aldolase; GAP: glyceraldehyde 3-phosphate; ED: Entner-Doudoroff; ns: not significant, that is BH-adj. p-value >0.01.

The online version of this article includes the following figure supplement(s) for figure 5:

**Figure supplement 1.** Genomic location and differential expression of genes encoding alginate catabolism and the flagellum locus.

these genes increased significantly on digested alginate relative to alginate (*Figure 4B*, *Figure 5A* and *Figure 5—figure supplement 1* and *Supplementary file 16*). Surprisingly, also the expression of most alginate lyase genes increased on digested alginate, especially the expression of secreted alginate lyase genes. This indicates the production of 'public' exoenzymes despite the abundance of monomeric and oligomeric degradation products in the digested alginate medium.

The increased motility of cells observed upon exposure to digested alginate (*Figure 3* and *Figure 4A*) led us to evaluate the expression of motility- and chemotaxis-associated genes across digested alginate and alginate treatments. As flagella are the main mode of motility in the genus *Vibrio* (*Khan et al., 2020*; *Echazarreta and Klose, 2019*), we focused on the expression of flagella-related genes. In the genome of *V. cyclitrophicus* ZF270 we found a gene cluster that encodes most genes involved in flagellar assembly and that was flanked by chemotaxis genes (hereon called flagellar locus, *Figure 4—figure supplement 1* and *Supplementary file 17*). Overall, 21/34 flagellar locus genes were differentially expressed (log2 fold change >0.5, BH-adjusted p-value <0.01) and the majority (90%) of these differentially expressed genes showed increased expression on digested alginate (*Figure 4B*). The flagellar genes *flgA*, *fliC*, and *fliH* and the chemotaxis gene *cheW* showed the strongest overexpression (5.4, 3.8, 2.0, and 2.6-fold, respectively). The expression of flagellar biosynthesis genes in *Vibrionacaea* occurs by a cascade of gene expression of four classes of genes (Class I - IV) (*Prouty et al., 2001*). We found the expression of the master regulator of the flagellar biosynthesis regulon, the Class I gene *flrA*, to be 1.6-fold increased on digested alginate (BH-adj. p-value 3e-19) (*Klose and Mekalanos, 1998*). The Class II regulatory genes *flrBC*, controlling Class III genes, and *fliA*, controlling Class IV genes, showed 1.5-fold and 1.3-fold increased expression on digested alginate (BH-adj. p-value 1e-12 for *flrB*, 2e-39 for *flrC*, 6e-15 for *fliA*; *Figure 5B*; *Echazarreta and Klose, 2019*; *Srivastava et al., 2013*). These findings suggest that the increased phenotypic motility observed on digested alginate (*Figure 3*) is related to the upregulation of flagellar biosynthesis genes. Additionally, the expression of the flagellum filament, encoded by *fliC* genes of the flagellar locus, was partially increased: *flaD* and *flaC* expression were increased by 3.8 and 1.6-fold, whereas the *flaA* gene was not significantly differentially expressed (BH-adj. *p*-value 7e-37, 2e-19, and 1e-02, respectively) (*Figure 5B* and *Supplementary file 6*). This suggests that cells grown on digested alginate have a *flaD*-rich filament composition, which has been shown to alter the swimming and adhesion characteristics of bacterial cells (*Kim et al., 2014*; *Nedeljković et al., 2021*). Lastly, we found that most genes involved in chemotaxis and located in the flagellar locus are highly expressed in cells grown on digested alginate (*Figure 4B* and *Figure 5C*) and likely drive the chemotactic activity of *V. cyclitrophicus* ZF270 during dispersal. Overall, our findings elucidate that cellular responses upon exposure to degradation products manifest in increased expression of genes involved in extracellular alginate breakdown and alginate catabolism as well as flagellar assembly and chemotaxis.

## Discussion

On Earth organic carbon is mostly present in the form of polysaccharides (*BeMiller, 2019*; *Reintjes et al., 2019*), which are often in a particulate state and form a heterogeneous resource landscape. Over the last years, the study of extracellular bacterial degradation of polysaccharides has revealed that bacterial growth on polysaccharides increases with increased cell density, enabling cells to benefit from the exoenzymes and extracellular degradation products of surrounding cells otherwise lost to diffusion ('cooperative growth'; *Ebrahimi et al., 2019*; *Drescher et al., 2014*; *Alcolombri et al., 2021*). However, cells have been observed not only to aggregate on polysaccharide sources, but also to leave them before the source is depleted (*Yawata et al., 2014*; *Alcolombri et al., 2021*). This cycle of biomass degradation and dispersal has long been discussed in the context of foraging e.g., *Yawata et al., 2014*; *Fenchel, 2002*; *Preheim et al., 2011*; *Yawata et al., 2020*; *McDougald et al., 2012*, but the cellular mechanisms that drive the cell dispersal remain unclear.

Our work links these observations and connects them to the cellular mechanisms that underlie the degradation-dispersal cycles of bacterial degraders, which we see as basal drivers of the biogeochemical processing of polysaccharides in heterogeneous nutrient-scapes. When bacteria encounter a new source of biomass, their local environment likely contains few mono- or oligosaccharides but is rich in polysaccharides which usually require extracellular breakdown (*Figure 6*, 'Finding a new nutrient source'). General concepts of how bacteria recognize the presence of large biopolymers remain elusive, but it was proposed that 'sentry' enzymes are constitutively expressed at a basal level to cleave mono- or oligosaccharides from polysaccharides, which cells can take up and which prime their metabolism for the degradation of the respective polysaccharide (*Thomas et al., 2012*; *Dudek et al., 2020*). It is not known yet how widespread the concept of sentry enzymes may be, but the observation of a constitutively expressed PL7 and PL15 family alginate lyase gene in *Z. galactanivorans* (*Thomas et al., 2012*) is mirrored in our work by the constant expression of an extracellular PL7

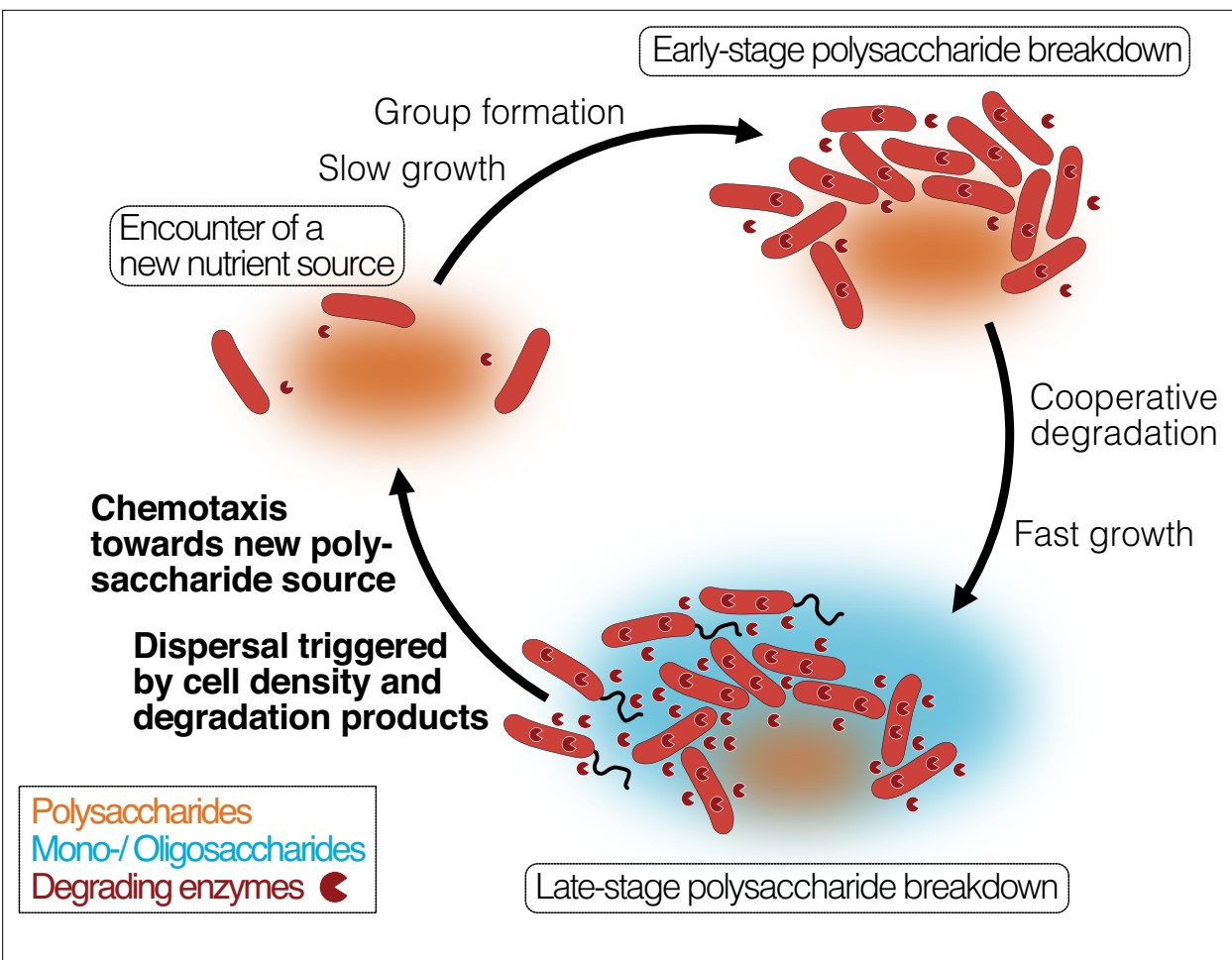

**Figure 6.** Bacterial growth and regulation on patches of polysaccharides. By integrating our results with previous studies on cooperative growth on the same system, as well as results on dispersal cycles in other systems, we highlight where the specific results of this work add to this framework (bold font). When cells encounter polymer sources, the colonization of the nutrient hotspot may be aided by the basal exoenzyme production of 'sentry' enzymes ('Encounter of a new carbon source'). This phase is succeeded by group formation, which enables cells to benefit from exoenzymes of neighboring cells and diffusing degradation products ('Early-stage polysaccharide breakdown'). The following phase includes cooperative extracellular degradation of the polysaccharide source, further increasing the concentration of available degradation products. These degradation products trigger the overexpression of alginate degrading, importing, and catabolizing enzymes, ensuring swift polysaccharide degradation ('Late-stage polysaccharide breakdown'). The increased pool of breakdown products also cues flagellar swimming in a subpopulation of cells and increases the expression of chemotaxis genes. Polymeric alginate acts as chemoattractant towards new polysaccharide sources. Cells and molecules are not drawn to scale. Dark red pie symbols: intracellular and extracellular polysaccharide-degrading enzymes; orange shading: a polymeric carbon source; blue shading: monomeric or oligomeric degradation products.

family alginate lyase gene, which may act as sentry enzyme that helps to initiate alginate degradation when cells encounter alginate.

Growth on polysaccharides has been found to be dependent on the cell density, as increased cell density limits the loss of degradation products and exoenzymes by diffusion (**D'Souza et al., 2023a**; **Figure 6**, 'Early-stage polysaccharide breakdown'). We found that cells grown on degradation products reach their maximal growth earlier and show increased expression of ribosomal biosynthesis, enzyme secretion, especially of secreted alginate lyases, transporters, quorum sensing and expression changes in the central carbon metabolism. The secretion of alginate lyases might seem surprising and wasteful in a monomer-rich environment. One reason for this observation may be that cells primarily rely on intracellular monosaccharide levels to trigger the upregulation of genes associated with polysaccharide degradation and catabolism, as has previously been observed for *E. coli* across various carbon sources (**Chubukov et al., 2014**; **Martínez-Antonio et al., 2006**). In fact, the majority of carbon sources are sensed by prokaryotes through one-component sensors

inside the cell (*Chubukov et al., 2014*). In the one-component internal sensing scheme, the enzymes and transporters for the use of various carbon sources are expressed at basal levels, which leads to an increase in pathway intermediates upon nutrient availability. The pathway intermediates are sensed by an internal sensor, usually a transcription factor, and lead to the upregulation of transporter and enzyme expression (*Chubukov et al., 2014*; *Martínez-Antonio et al., 2006*). This results in a positive feedback loop, which enables small changes in substrate abundance to trigger large transcriptional responses (*Chubukov et al., 2014*; *Wall et al., 2004*). Thus, the presence of alginate breakdown products may likely result in increased expression of all components of the alginate degradation pathway, including the expression of degrading enzymes. As the gene expression analysis was performed on well-mixed cultures in culture medium containing alginate breakdown products, we therefore expect a strong stimulation of alginate catabolism. In a natural scenario, where cells disperse from a polysaccharide hotspot before its exhaustion, the expression of alginate catabolism genes may likely decrease again once the local concentration of breakdown products decreases. However, continued production of alginate lyases could also provide an advantage when encountering a new alginate source and continued production of alginate lyases may thus help cells to prepare for likely future environments. Further investigations of bacterial enzyme secretion in changing nutrient environments and at relevant spatial scales are required to improve our understanding of the regulation of enzyme secretion along nutrient gradients.

We show that cells respond to the exposure to degradation products with dispersal from dense cell groups by means of increased flagellum-driven swimming (*Figure 6*, 'Late-stage polysaccharide breakdown'), decreasing their local cell number. This finding matches with previous observations of cells leaving biopolymer particles before they are depleted (*Yawata et al., 2014*; *Alcolombri et al., 2021*). A plausible explanation for this density-dependent dispersal is that cells in larger groups compete with each other for nutrients and space, while not profiting from cooperative degradation anymore due to the abundance of degradation products. Motility has also been shown to increase the encounter rate of cells with sources of nutrients (*Bassler et al., 1991*; *Meibom et al., 2004*), suggesting motility as a strategy that allows cells to escape from the ensuing competition. Previous work in *Caulobacter crescentus* demonstrated that a flagellum knock-out mutant formed larger cell groups, resulting in reduced growth rates due to intercellular competition (*Povolo et al., 2022*; *D'Souza et al., 2021*). While it would be interesting to study non-motile mutants of *V. cyclitrophicus* ZF270, the non-tractability of natural isolates makes direct tests of molecular mechanisms difficult. Additionally, subjecting cells separately to the two monomeric units of alginate or oligomers of defined size could improve our understanding of the specific molecules that trigger motility, but this was experimentally not feasible.

Direct chemotaxis towards polysaccharides may facilitate the search for new polysaccharide sources after dispersal. We found that the presence of degradation products not only induces cell dispersal but also increases the expression of chemotaxis genes. Interestingly, we found that *V. cyclitrophicus* ZF270 cells show chemotaxis towards polymeric alginate but not digested alginate. This contrasts with previous findings for bacterial strains degrading the insoluble marine polysaccharide chitin, where chemotaxis was strongest towards chitin oligomers (*Bassler et al., 1991*), suggesting that oligomers may act as an environmental cue for polysaccharide nutrient sources (*Keegstra et al., 2022*). However, recent work has shown that certain marine bacteria are attracted to the marine polysaccharide laminarin, and not laminarin oligomers (*Clerc et al., 2023*). Together with our results, this indicates that chemotaxis towards soluble polysaccharides may be mediated by the polysaccharide molecules themselves. The mechanism of this behavior is yet to be identified, but could be mediated by polysaccharide-binding proteins as have been found in *Sphingomonas* sp. A1 facilitating chemotaxis towards pectin (*Konishi et al., 2020*). Direct polysaccharide sensing adds complexity to chemosensing as polysaccharides cannot freely diffuse into the periplasm, which can lead to a trade-off between chemosensing and uptake (*Norris et al., 2022*). Furthermore, most polysaccharides are not immediately metabolically accessible as they require degradation. But direct polysaccharide sensing can also provide certain benefits compared to using oligomers as sensory cues. First, it could enable bacterial strains to preferably navigate to polysaccharide nutrients sources that are relatively uncolonized and hence show little degradation activity. Second, strong chemotaxis towards degradation products could hinder a timely dispersal process as the dispersal then requires cells to travel against a strong attractant gradient formed by the degradation products. Overall, this strategy allows cells to alternate between degradation and dispersal to acquire carbon and energy in a heterogeneous

world with nutrient hotspots (*Fenchel, 2002*; *Blackburn and Fenchel, 1999*; *Blackburn et al., 1998*; *Smriga et al., 2016*).

## Conclusion

The heterogeneous landscape of polysaccharide hotspots in natural systems requires bacteria to effectively break down polymeric carbohydrates as well as readily ensure dispersal to new nutrient hotspots. Our findings show that the degree of depolymerization of the polysaccharide influences this decision, altering the growth dynamics, metabolic activity, and motility of cells. Our study also contextualizes the surprising finding that foraging bacteria majorly leave polysaccharide particles before the last third of the particle is consumed (*Alcolombri et al., 2021*). Dispersal from a partially degraded carbon source may serve several purposes: (i) escaping competition that ensues within large cell groups, (ii) ensuring the spread of a part of the clonal population to new environments as bet hedging strategy (*McDougald et al., 2012*; *Ronce, 2007*), here guided by chemotaxis towards new nutrient hotspots, (iii) preventing whole populations degrading a sinking marine particle or a deposited sediment particle to be buried in depth where nutrient hotspots become sparse (*Alcolombri et al., 2021*; *Fenchel, 2008*), and/or (iv) increase the genetic variation in bacterial populations (*McDougald et al., 2012*). However, dispersal may also occur when a nutrient source offers a surplus of carbon while other essential nutrients become limiting, as the increased expression of iron, zinc, and phosphate transporters in cells grown on digested alginate suggested. These findings emphasize that metabolic molecules can also act as triggers of dispersal, expanding upon the current perspective of dispersal in biofilms as a reaction to dispersal cues like NO, signaling molecules, nutrient starvation, and oxygen starvation (*Rumbaugh and Sauer, 2020*). The study of bacterial motility on increasingly complex biomass particles will reveal the role of the nutrient composition of the present nutrient hotspot on the bacterial decision-making. Overall, these new insights into the cellular mechanisms and regulation that drive degradation-dispersal cycles contribute to our understanding of the microbially driven remineralization of biomass, and factors that modulate this process. The open questions of how bacteria sense polysaccharides in their environment, which cell signaling pathways integrate the presence of degradation products in the cellular decision-making of degradation and dispersal, and to what extent cell populations coordinate this decision, present an exciting avenue of further research.

## Materials and methods

### Bacterial strains, media, and growth assays

*Vibrio cyclitrophicus* ZF270 (available through Culture Collection Of Switzerland; Accession number: 2043) cells were cultured in Marine Broth (DIFCO) and grown for 18 hr at 25 °C. Cells from these cultures were used for growth experiments in Tibbles Rawling (TR) salts minimal medium (*Hehemann et al., 2016*; *Tibbles and Rawlings, 1994*) containing either 0.1% (weight/volume) algae-derived alginate (referred to as 'alginate'; Sigma-Aldrich, CAS-number 9005-38-3) or 0.1% (weight/volume) digested alginate. At these concentrations, both alginate and digested alginate are soluble in the culture medium. The digested alginate was produced by enzymatically digesting 2% alginate with 1 unit ml$^{-1}$ of alginate lyase (Sigma-Aldrich, CAS-number 9024-15-1) at 37 °C for 48 hr. In our experiment we used 1 unit/ml of alginate lyases in a 4.5 ml solution to digest the alginate. As the commercially purchased alginate lyases are 10,000 units/g, our 4.5 ml solution contains 0.45 mg of alginate lyase protein. The digested alginate solution diluted 45 x when added to culture medium. This means that we added 0.18 µg alginate lyase protein to 1 ml of culture medium. Based on the above calculation, we conclude that the amount of protein added to the growth medium by the addition of alginate lyases is so small that we consider it negligible. As a comparison, for 1 ml of alginate medium, 1000 µg of alginate is added or for 1 ml of Lysogeny broth (LB) culture medium, 3,500 µg of LB are added. Thus, the amount of alginate lyase protein that we added is ca. 5000–20,000 times smaller than the amount of alginate or LB that one would add to support cell growth. Therefore, we expect the growth that the digestion of the added alginate lyases would allow to be negligible.

Carbon sources were prepared in nanopure water and filter sterilized using 0.40 µm Surfactant-Free Cellulose Acetate filters (Corning, USA). Well-mixed batch experiments in alginate or digested alginate medium were performed in 96-well plates (Greiner Bio) and growth dynamics were measured using a microwell plate reader (Biotek, USA). Plate reader assays were initiated as described previously

(**D'Souza et al., 2014**). Briefly, 1 ml from a culture grown for 18 hr on Marine broth was centrifuged at 5000 × *g* in 1.5 ml microfuge tubes for 5 min. The supernatant was discarded and the cell pellet was subjected to two rounds of washing with the basal TR salts medium. The cell-pellet was then resuspended in 1 ml of TR salts medium and 5 µl of this suspension inoculated into 195 µl TR medium with either carbon source ~10^5 colony forming units (CFUs ml^{-1}) in a 96-well plate (Greiner Bio). The optical density (600 nm) was then measured every 15 min for 40 hours. All measurements had six biological replicates.

## Alginate oligosaccharide measurements

Oligosaccharide measurements were performed using liquid chromatography time of flight mass spectrometry (LC-QTOF-MS). Samples were prepared by diluting 1:20 in milliQ water and 5 µL of sample was injected per measurement. Chromatographic separation was performed using an Agilent 1290 stack, using an Agilent HILIC-Z column (2.7 µm particles, 2.1x50 mm). Mobile phase A contained 10% acetonitrile (Fisher Scientific) and 0.1% medronic acid (Agilent), and Mobile phase B contained 90% acetonitrile and 0.1% medronic acid. The separation was performed as follows: Mobile phase B 100% for 1 min, gradient to 30% phase B over 3 min, 30% phase B for 30 s, and equilibration of 100% phase B for 5 min. The flow rate was 400 µL min^{-1} at 30 °C. Samples were measured using an Agilent 6520 mass spectrometer in negative mode, in 4 GHz high-resolution mode. Data analysis was performed in Agilent Quantitative Analysis software.

## Microfluidics and time-lapse microscopy

Microfluidic experiments and microscopy were performed as described previously (**Dal Co et al., 2020**; **D'Souza et al., 2021**; **Mathis and Ackermann, 2016**). Cells were imaged within chambers of a PDMS (Sylgard-Dow) microfluidic chip that ranged in size from 60 to 120×60 × 0.56 µm (*l*×*b* × h). Within these chambers, cells can attach to the glass surface and experience the medium that diffuses through lateral flow channels. Imaging was performed using IX83 inverted microscope systems (Olympus, Japan) with automated stage controller (Marzhauser Wetzlar, Germany), shutter, and laser-based autofocus system (Olympus ZDC 2). Chambers were imaged in parallel on the same PDMS chip, and phase-contrast images of each position were taken every 8 or 10 min. The microscopy unit and PDMS chip were maintained at 25 °C using a cellVivo microscope incubation system (Pecon GmbH).

## Viscosity of the alginate and digested alginate solution

We measured the viscosity of alginate solutions using shear rheology measurements. We use a 40 mm cone-plate geometry (4° cone) in a Netzsch Kinexus Pro +rheometer. A total of 1200 µL of sample was placed on the bottom plate, the gap was set at 150 µm and the sample trimmed. We used a solvent trap to avoid sample evaporation during measurement. The temperature was set to 25 °C using a Peltier element. We measure the dynamic viscosity over a range of shear rates = 0.1–100 s-1. We report the viscosity of each solution as the average viscosity measured over the shear rates 10–100 s-1, where the shear-dependence of the viscosity was low.

We measured the viscosity of 0.1% (w/V) alginate dissolved in TR media, which was 1.03+/-0.01 mPa·s (reporting the mean and standard deviation of three technical replicates.). The viscosity of 0.1% digested alginate in TR media was found to be 0.74+/-0.01 mPa·s. This means that the viscosity of alginate in our microfluidic experiments is 36% higher than of digested alginate, but the viscosities are close to those expected of water (0.89 mPa·s at 25°C according to **Berstad et al., 1988**).

## Motility assays

Cells were grown for 10 hr in Marine Broth (DIFCO) after which 10 µl of culture was used to inoculate culture tubes (Greiner) containing 5 ml of TR medium with either 0.1% alginate or 0.1% digested alginate. After 6 hr of growth at 25 °C, 2 µl of cell suspension was inoculated into microfluidic growth chambers. Cells within six replicate chambers were then imaged with the phase-contrast channel at a high frame rate (125 Hz, i.e. frames s^{-1}) using the same microscopy setup described above.

## Chemotaxis assays

To assess whether polymeric alginate and digested alginate attract *Vibrio cyclitrophicus* ZF270, we used the In Situ Chemotaxis Assay (**Lambert et al., 2017**; **Clerc et al., 2020**; ISCA), a microfluidic

device consisting of a 5×5 array of microwells that can be individually loaded with solutions of different chemicals (110 µl each). Once the ISCA is deployed in an aqueous environment, the chemicals diffuse out of the wells through a small port, creating chemical gradients which will guide chemotactic bacteria inside the wells of the device (*Lambert et al., 2017*; *Clerc et al., 2020*). *Vibrio cyclitrophicus* ZF210 was plated on Marine Agar (BD Difco) from a glycerol stock and grown for 16 hr at 27 °C. A single colony was then incubated in 10% Marine Broth (BD Difco) in 0.22 µm filtered artificial seawater (Instant Ocean, Spectrum Brands) and grown overnight at 27 °C and 180 rpm. The culture was diluted down to $1 \times 10^6$ cells ml$^{-1}$ in 0.22 µm filtered artificial seawater (Instant Ocean, Spectrum Brands) to perform the chemotaxis experiment. Both chemoattractants (alginate and digested alginate) were diluted in sterile seawater (35 g l$^{-1}$; Instant Ocean, Spectrum Brands) at a final concentration of 0.1% and then filtered with a 0.2 µm filter (Millipore) to remove particles and potential contaminants. Within the ISCA, one full row of five wells was used per chemoattractant as technical replicates. The chemoattractants were injected in triplicate ISCA with a sterile 1 ml syringe (Codau) and needle (27 G, Henke Sass Wolf). A last row containing 0.2 µm-filtered seawater acted as negative control accounting for cells swimming in the device by random motility only. Experiments were conducted by incubating the ISCAs for 1 hr in the diluted *Vibrio cyclitrophicus* ZF270 culture. Upon time completion, a sterile syringe and needle were used to retrieve the content of the wells and transferred to 1 ml microfuge tubes resulting in a pooling of a row of five wells containing the same sample. Sample staining was performed with SYBR Green I (Thermo Fisher) and the chemotactic response was quantified by counting cells using flow cytometry. The strength of the chemotactic response was determined by the mean chemotactic index ($I_C$), defined as the ratio of the number of cells found in each chemoattractant to the number of cells in control wells containing filtered seawater (so that attraction corresponds to $I_C > 1$).

## Culturing and harvesting cells for transcriptomics

Cells were grown for 18 hr in Marine Broth (DIFCO) after which 1 ml of culture was centrifuged at $5000 \times g$ in 1.5 ml microfuge tubes for 5 min. The supernatant was discarded and the cell pellet was subjected to two rounds of washing with the basal TR salts medium. The cell pellet was then resuspended in 1 ml of TR salts medium and 250 µl of this suspension were used to inoculate 100 ml flasks (Schott-Duran) containing 10 ml of TR medium with either 0.1% alginate or 0.1% digested alginate. This was done in parallel for six flasks. Once cultures in the flasks reached mid-exponential phase (10 hr and 15 hr after inoculation for digested alginate and alginate, respectively) and had approximately the same OD (0.39 for digested alginate and 0.41 for alginate), 2 ml of cultures were harvested for RNA extraction. Samples were stabilized with the RNprotect reagent (Qiagen) and RNA was extracted using the RNeasy mini kit (Qiagen).

## Sequencing and gene annotation of *Vibrio cyclitrophicus* ZF270

Long read sequencing using the Oxford Nanopore Platform (Long read DNA sequencing kit) and short read sequencing using the Illumina platform (Illumina DNA Prep kit and IDT 10 bp UDI indices, and sequenced on an Illumina NextSeq 2000 producing 2x151 bp reads) was performed by the Microbial Genome Sequencing Center, Pittsburgh, USA (MiGS), to create a new closed reference genome of *V. cyclitrophicus* ZF270 (BioProject PRJNA991487). Annotation of this genome was done with RASTtk (v2.0, Rapid Annotation using Subsystem Technology tool kit *Brettin et al., 2015*). Additionally, KEGG Ontology identifiers ('K numbers') were annotated with BlastKOALA (v2.2; *Kanehisa et al., 2016*). Dedicated annotation of alginate lyase genes was performed by homology search for proteins belonging to the PL5, PL6, PL7, PL14, PL15, PL17, PL18, PL31, PL36, or PL39 family *Cheng et al., 2020* by dbCAN2 (v9.0) (*Zhang et al., 2018*). Enzymes for alginate transport and metabolism were identified by BLASTn-search *Coordinators, 2016*; *Altschul et al., 1990* of gene sequences of *Vibrio splendidus* 12B01, which were previously identified as minimum genetic prerequisites for alginate utilization and enabled alginate degradation when cloned into *E. coli* (*Wargacki et al., 2012*).

## Location prediction of alginate lyases

Signal peptides were annotated using SignalP (v.5.0) *Almagro Armenteros et al., 2019*, and LipoP (v.1.0) (*Juncker et al., 2003*). SignalP discriminated between (1) Sec/SPI: 'standard' secretory signal peptides transported by the Sec translocon and cleaved by Signal Peptidase I (SPI), (2) Sec/SPII: lipoprotein signal peptides transported by the Sec translocon and cleaved by Signal Peptidase II

(SPII), and (3) Tat/SPI: Tat signal peptides transported by the Tat translocon and cleaved by SPI. LipoP discriminates between (1) SPI: signal peptide, (2) SpII: lipoprotein signal peptide, and (3) TMH: n-terminal transmembrane helix. All predictions were in agreement, apart from one PL7 (gene 1136176.5. peg.4375) which was predicted by LipoP as cytoplasmic and by SignalP as equally likely cytoplasmic as containing a lipoprotein signal peptide.

## Transcriptomic analysis: Sequencing, pre-processing, differential expression analysis, and functional analysis

Sequencing (12 M reads, 2x50 bp) of the isolated RNA was performed by MiGS after rRNA depletion using RiboZero Plus (Ilumina). cDNA libraries were prepared using an Illumina DNA Prep kit and IDT 10 bp UDI indices, and sequenced on an Illumina NextSeq 2000. Preprocessing of the raw reads was carried out as follows: Quality control was performed with FastQC (v0.11.9) *Andrews, 2010* and reads were trimmed with Trimmomatic (v0.38) *Bolger et al., 2014*; the high-quality reads were mapped to the reference genome (described above) with Bowtie2 (v2.3.5.1) *Langmead and Salzberg, 2012*; binarization, sorting, and indexing were done with Samtools (v1.10) *Danecek et al., 2021*; gene counts were computed with the featureCount function of Subread (v2.0.1) (*Liao et al., 2014*).

Differential expression analysis was performed with DESeq2 (v1.30.1) (*Love et al., 2014*). In brief, DESeq2 normalizes the raw read counts with normalization factors ('size factors') to account for differences in sequencing depth between samples. Subsequently, gene-wise dispersion estimates are computed for each gene separately using maximum likelihood, and then shrunk toward the values predicted by the dispersion-mean dependence curve to obtain final dispersion values. Finally, DESeq2 fits a negative binomial model to the read counts and performs significance testing using the Wald test. Here reported p-values result from the Wald test of read counts from the digested alginate condition compared to the alginate condition and were adjusted for multiple testing by Benjamini-Hochberg correction (*Benjamini and Hochberg, 1995*) as implemented in the p.adjust function of base R (v4.1.2). The reported log2 fold changes indicate the log2(DESeq2-normalised reads in digested alginate condition / DESeq2-normalised reads in alginate condition) for each gene.

Visualization of gene maps was performed in R with the ggplot2 package (v3.4.0) and the extension gggenes (v0.4.1) by David Wilkins.

For systematic functional analysis we performed gene set enrichment analysis (GSEA) (*Subramanian et al., 2005*) using the fgsea function of the fgsea package (v1.20.0) with minimal number of unique genes per gene set 'minSize'=5 and number of permutations 'nPermSimple'=1000000. In brief, GSEA takes the full gene list ranked by log2 fold change and annotated with K numbers as input and determines whether the member genes of any KEGG pathway are randomly distributed throughout the ranked gene list or whether they are primarily found at the top or bottom (*Subramanian et al., 2005*). This is quantified by the enrichment score (ES), which corresponds to a weighted Kolmogorov-Smirnov-like statistic. The ES of each gene set is normalized to the mean enrichment of random samples of the same size to account for the size of the set, yielding the normalized enrichment score (NES). To estimate the significance level of the enrichment score, the p-value of the observed enrichment score is calculated relative to a null distribution that was computed from permuted data. The estimated significance level was adjusted to account for multiple hypothesis testing. As gene sets we chose all KEGG pathways and KEGG BRITE categories (as noted in *Supplementary file 1* in column 'KEGG_pathway') within all genes of *V. cyclitrophicus* ZF270 annotated with a KEGG Ontology identifier ('K number'). Visualization of differential expression levels in KEGG pathways was performed with the R package pathview (v1.35.0) (*Luo and Brouwer, 2013*). We also formed gene sets of the genes associated with alginate utilization (see '*Sequencing and gene annotation of Vibrio cyclitrophicus ZF270*', *Supplementary file 16*), of the genes of the flagellar locus that map to the KEGG pathways 'Bacterial motility proteins', and of the genes of the flagellar locus that map to the KEGG pathways 'Bacterial chemotaxis' (as noted in *Supplementary file 17*) within all genes of *V. cyclitrophicus* ZF270.

## Image analysis

Cells within microscopy images were segmented and tracked using ilastik (v1.3) ('pixel classification workflow' and 'tracking with probabilities workflow'). Phase contrast images were used for alignment, segmentation, tracking and linking. Images were cropped at the boundaries of each microfluidic chamber. The lineage identity of each single cell was assigned by ilastik's tracking plugin and

visualized by coloring the segmented cells, respectively. The growth rate of each cell was computed as the change of cell area over time, that is via a linear regression of the single-cell area over the time between consecutive cell divisions, based on ilastik's segmentation and tracking output. Cells that were tracked over less than three frames were excluded. Measurement of swimming speeds and displacement of cells was performed using ilastik (v1.4), ImageJ (v2.3) and Trackmate (v7.5.2). Briefly, cell-segmentation ('pixel classification workflow' in ilastik) and tracking ('animal tracking workflow' in ilastik) were performed using the high frame rate phase contrast images in ilastik (v1.4). Cell trajectories and properties were then computed using the output of the ilastik workflow in Trackmate.

### Dispersal analysis

For the analysis on the dispersal of cells (*Figure 2*), we computed the cell number as the total number of cells within a microfluidic chamber. The change in the number of cells was computed by subtracting the number of cells before the medium switch (i.e. average number of cells between t=1.9–2.1 hr) from the number of cells after the medium switch (i.e. average number of cells between t=3.9–5.5 hr).

### Datasets and statistical analysis

All batch experiments were replicated three to six times. Growth curves were analyzed in Python (v3.7) using the *Amiga* package (v1.1.0) *Midani et al., 2021* and GraphPad Prism (v8, GraphPad Software). The microscopy dataset consisted of eight chambers each, corresponding to the eight replicates shown in *Figure 1* and *Figure 2*. These were grouped into two biological replicates wherein each biological replicate was fed by media through a unique channel in a microfluidic chip. Cells with negative growth rates were excluded from the analysis after visual curation, as they represented artifacts, mistakes in segmentation or linking during the tracking process, or non-growing deformed cells. Each chamber was treated as an independent replicate. Comparisons were considered statistically significant when $p<0.05$ or when the False Discovery Rate (FDR)-corrected $q$ was smaller than 0.05. FDR corrections were applied when multiple t tests were performed for the same dataset. Measures of effect size are represented by the $R^2$ or eta (*Reintjes et al., 2019*) value. All statistical analyses were performed in GraphPad Prism v9.0 (GraphPad Software, USA), R v4.1.2, RStudio v1.1.463 (Posit, USA).

## Acknowledgements

We thank past and present members of the Microbial Systems Ecology group for feedback. This research was supported by an ETH fellowship and a Marie Skłodowska-Curie Actions for People COFUND program fellowship (FEL-37-16-1) to GD; an ETH Career Seed Grant to GD (FEL-14 18–1), the Simons Foundation Collaboration on Principles of Microbial Ecosystems (PriME, #542379 and #542395) to MA and RS; and by ETH Zurich and Eawag.

---

## Additional information

### Funding

| Funder | Grant reference number | Author |
|---|---|---|
| Marie Sklodovska-Curie Actions for People COFUND program fellowship | FEL-37-16-1 | Glen G D'Souza |
| ETH Zurich | ETH Career Seed Grant FEL-14 18-1 | Glen G D'Souza |
| Simons Foundation | #542379 | Martin Ackermann |
| Simons Foundation | #542395 | Roman Stocker |

The funders had no role in study design, data collection and interpretation, or the decision to submit the work for publication.

## Author contributions

Astrid Katharina Maria Stubbusch, Conceptualization, Data curation, Formal analysis, Investigation, Methodology, Project administration, Validation, Visualization, Writing – original draft, Writing – review and editing; Johannes M Keegstra, Conceptualization, Formal analysis, Funding acquisition, Investigation, Methodology, Visualization, Writing – review and editing; Julia Schwartzman, Conceptualization, Funding acquisition, Writing – review and editing; Sammy Pontrelli, Estelle E Clerc, Investigation, Methodology, Writing – review and editing; Samuel Charlton, Investigation; Roman Stocker, Funding acquisition, Writing – review and editing; Cara Magnabosco, Conceptualization, Resources, Supervision, Writing – review and editing; Olga T Schubert, Martin Ackermann, Conceptualization, Funding acquisition, Project administration, Resources, Supervision, Writing – review and editing; Glen G D'Souza, Conceptualization, Data curation, Formal analysis, Funding acquisition, Investigation, Methodology, Project administration, Validation, Visualization, Writing – review and editing

## Author ORCIDs

Astrid Katharina Maria Stubbusch ⓘ https://orcid.org/0000-0003-4767-2712
Johannes M Keegstra ⓘ https://orcid.org/0000-0002-8877-4881
Julia Schwartzman ⓘ https://orcid.org/0000-0003-4563-4835
Olga T Schubert ⓘ https://orcid.org/0000-0002-2613-0714
Martin Ackermann ⓘ https://orcid.org/0000-0003-0087-4819
Glen G D'Souza ⓘ https://orcid.org/0000-0002-9123-101X

Reviewer #1 (Public review): https://doi.org/10.7554/eLife.93855.3.sa1
Reviewer #2 (Public review): https://doi.org/10.7554/eLife.93855.3.sa2
Reviewer #3 (Public review): https://doi.org/10.7554/eLife.93855.3.sa3
Author response https://doi.org/10.7554/eLife.93855.3.sa4

## Additional files

### Supplementary files

• Supplementary file 1. Differential gene expression of all genes of *V. cyclitrophicus* ZF270. Genes of *V. cyclitrophicus* ZF270 were annotated by RASTtk. Differential expression analysis was performed on all genes with DESeq2 v1.32.0 *Bassler et al., 1991* to compute the log2 fold change in gene expression for each gene and the corresponding *p*-value by Benjamini-Hochberg-adjusted Wald test. Geneid: gene identifier of genome annotation file; Chr: chromosome; Start: start of gene in base pairs; End: end of gene in base pairs; Strand: DNA strand on which gene is located; Length: length of gene; L1_raw to L6_raw: raw read count of replicate 1–6 on digested alginate; P1_raw to P6_raw: raw read count of replicate 1–6 on polymeric alginate; L1_DESeq to L6_DESeq: DESeq2-normalized read count of replicate 1–6 on digested alginate; P1_DESeq to P6_DESeq: DESeq2-normalized read count of replicate 1–6 on polymeric alginate; baseMean: baseMean value computed with DESeq2; log2FoldChange: log2 fold change value computed with DESeq2; lfcSE: shrunken (posterior) standard deviation computed with DESeq2; stat: Wald statistic computed with DESeq2, i.e. the log2 fold change divided by lfcSE, which is compared to a standard Normal distribution to generate a two-tailed *p*-value; pvalue: Wald test *p*-value computed with DESeq2; padj: Benjamini-Hochberg-adjusted Wald test *p*-value computed with DESeq2; RASTtk_Annotation: gene annotation by RASTtk; RASTtk_Ontology_term: ontology term by RASTtk; BlastKOALA_KO: KEGG Orthology by BlastKOALA; BlastKOALA_KO_Definition: KEGG Orthology definition by BlastKOALA; BlastKOALA_KO_Score: weighted sum of BLAST bit scores computed by BlastKOALA; KEGG_pathway: ID of the KEGG category C associated with the KEGG Orthology (BlastKOALA_KO), i.e., KEGG pathway ID or KEGG BRITE ID; KEGG_pathway_descr: Description of the KEGG category C; KEGG_CategB: ID of the KEGG category B associated with the KEGG Orthology (BlastKOALA_KO); KEGG_CategA: ID of the KEGG category A associated with the KEGG Orthology (BlastKOALA_KO). All KEGG categories were based on https://www.kegg.jp/kegg-bin/show_brite?ko00001.keg , Mar 18 2021.

• Supplementary file 2. Genome-wide pathway enrichment analysis. Performed on all gene sets of KEGG category C (KEGG pathways and KEGG BRITE categories) by Gene Set Enrichment Analysis algorithm (GSEA) (*Prouty et al., 2001*). Method: fgsea() function and described filtering (see Materials and Methods); KEGG_hierarchy: ID of KEGG category C; KEGG_entry: KEGG pathway or KEGG BRITE category; Description: Description of the KEGG category; pval: enrichment *p*-value of GSEA; padj: BH-adjusted *p*-value of GSEA; log2err: the expected error for the standard deviation

of the *p*-value logarithm, ES: enrichment score, same as in Broad GSEA implementation; NES: normalized enrichment score, normalized to mean enrichment of random samples of the same size; size: size of gene set after removing genes not present in the genome of *V. cyclitrophicus* ZF270; Genes_total_ZF270: number of genes of *V. cyclitrophicus* ZF270 within the gene set, counting gene duplicates.

• Supplementary file 3. Differential expression in genes of the valine, leucine and isoleucine biosynthesis (a subset of *Supplementary file 1*).

• Supplementary file 4. Differential expression in genes of the propanoate metabolism (a subset of *Supplementary file 1*).

• Supplementary file 5. Differential expression in genes encoding the ribosome (a subset of *Supplementary file 1*).

• Supplementary file 6. Differential expression in genes of the secretion system (a subset of *Supplementary file 1*).

• Supplementary file 7. Differential expression in genes of the bacterial secretion system (a subset of *Supplementary file 1*).

• Supplementary file 8. Differential expression in genes of the general secretion pathway (a subset of *Supplementary file 1*).

• Supplementary file 9. Differential expression in genes of enzymes with EC numbers (a subset of *Supplementary file 1*).

• Supplementary file 10. Differential expression in genes of transporters (a subset of *Supplementary file 1*).

• Supplementary file 11. Differential expression in genes of ABC transporters (a subset of *Supplementary file 1*).

• Supplementary file 12. Differential expression in genes of the bacterial motility proteins (a subset of *Supplementary file 1*).

• Supplementary file 13. Differential expression in genes associated with quorum sensing (a subset of *Supplementary file 1*).

• Supplementary file 14. Differential expression in genes of transcription factors (a subset of *Supplementary file 1*).

• Supplementary file 15. Differential expression in genes of beta-Lactam resistance (a subset of *Supplementary file 1*).

• Supplementary file 16. Differential expression of alginate lyases (*PL6, PL7, PL15, PL17*), transporters (porin *kdgM*, symporter *toaB*, symporter *toaC*), and metabolic enzymes shunting into the ED pathway (DEHU reductase *DehR, kdgK, eda*) (a subset of *Supplementary file 1*).

• Supplementary file 17. Differential expression of genes of the flagellum locus, comprising the cluster of genes that was part of the KEGG category of bacterial motility (a subset of *Supplementary file 1*).

• MDAR checklist

### Data availability

Sequencing data have been deposited on NCBI, BioProject PRJNA991487. All further data and code is deposited on ERIC Open (https://opendata.eawag.ch) at https://doi.org/10.25678/0008MH.

The following datasets were generated:

| Author(s) | Year | Dataset title | Dataset URL | Database and Identifier |
|---|---|---|---|---|
| Stubbusch AKM, Keegstra JM, Schwartzman J, Pontrelli S, Clerc EE, Charlton S, Stocker R, Magnabosco C, Schubert OT, Ackermann M, D'Souza GG | 2024 | Data from: Vibrio cyclitrophicus ZF270 on alginate and digested alginate | https://www.ncbi.nlm.nih.gov/datasets/genome/GCF_038442155.1 | NCBI Genome, GCF_038442155.1 |
| Stubbusch AKM, Keegstra JM, Schwartzman J, Pontrelli S, Clerc EE, Charlton S, Stocker R, Magnabosco C, Schubert OT, Ackermann M, D'Souza GG | 2024 | Vibrio cyclitrophicus ZF270 on alginate and digested alginate | https://www.ncbi.nlm.nih.gov/bioproject/PRJNA991487 | NCBI BioProject, PRJNA991487 |
| Stubbusch AKM, Keegstra JM, Schwartzman J, Pontrelli S, Clerc EE, Charlton S, Stocker R, Magnabosco C, Schubert OT, Ackermann M, D'Souza GG | 2024 | Data for: Polysaccharide breakdown products drive degradation-dispersal cycles of foraging bacteria through changes in metabolism and motility | https://doi.org/10.25678/0008MH | ERIC Open, 10.25678/0008MH |

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
