## [Editor Report · eLife assessment]

This manuscript is a **valuable** contribution to our understanding of foraging behaviors in marine bacteria. The authors present a conceptual model for how a marine bacterial species consumes an abundant polysaccharide. Using experiments in microfluidic devices and through measurements of motility and gene expression, the authors offer **convincing** evidence that the degradation products of polysaccharide digestion can stimulate motility.

---

## [Referee Report · Reviewer #1 (Public review)]

Summary:

The authors attempt to understand how cells forage for spatially heterogeneous complex polysaccharides. They aimed to quantify the foraging behavior and interrogate its genetic basis. The results show that cells aggregate near complex polysaccharides and disperse when simpler byproducts are added. Dispersing cells tend to move towards the polysaccharide. The authors also use transcriptomics to attempt to understand which genes support each of these behaviors - with motility and transporter related genes being highly expressed during dispersal, as expected.

Strengths:

The paper is well written and builds on previous studies by some of the authors showing similar behavior by a different species of bacteria (Caulobacter) on another polysaccharide (xylan). The conceptual model presented at the end encapsulates the findings and provides an interesting hypothesis. I also find the observation of chemotaxis towards the polysaccharide in the experimental conditions interesting.

Weaknesses:

Much of the genetic analysis, as it stands, is quite speculative and descriptive. I found myself confused about many of the genes (e.g., quorum sensing) that pop up enriched during dispersal quite in contrast to my expectations. While the authors do discuss this in the text as worth following up on, I think the analysis as it stands is speculative about the behaviors observed. In the authors' defense, I acknowledge that it might have the potential to generate hypotheses and thus aid future studies.

---

## [Referee Report · Reviewer #2 (Public review)]

Summary:

The paper sets out to understand the mechanisms underlying the colonization and degradation of marine particles using a natural Vibrio isolate as a model. The data are measurements of motility and gene expressing using microfluidic devices and RNA sequencing. The results reveal that degradation products of alginate do stimulate motility but not chemotaxis. In contrast, alginate itself (the polymer) does stimulate chemotaxis. Further, the dispersal from degrading alginate is density dependent, increasing at higher density. The evidence for these claims are strong. From these the authors propose a narrative (Fig. 6) for growth and dispersal cycles in this system. The idea is that cells colonize and degrade alginate, this degradation stimulates motility and dispersal followed by chemotaxis to a new alginate source. This complete narrative has modest support in the data. A quantitative description of these dynamics awaits future studies.

Strengths:

The microfluidic measurements are the central strength of the paper. The density dependence claim is qualitatively supported by the data. The motility and chemotaxis claims are also well supported by the data. The presentation of the experiment and results are well done. The study serves to motivate a unifying picture of growth and dispersal in marine systems. This is a key process in the global carbon cycle.

Weaknesses:

Perhaps not a weakness, but a glimmer that this is not yet the full story. The RNA expression data show alginate lyase expression in response to digested alginate which is unexpected given the narrative articulated above. Why express lyases while leaving the polymer patch via motility? This question is addressed in the Discussion. A holistic and quantitative picture of the proposed process in Figure 6 awaits additional studies.

---

## [Referee Report · Reviewer #3 (Public review)]

Summary:

In this manuscript, Stubbusch and coauthors examine the foraging behavior of a marine species consuming an abundant marine polysaccharide. Laboratory experiments in a microfluidic setup are complemented with transcriptomic analyses aiming at assessing the genetic bases of the observed behavior. Bacterial cells consuming the polysaccharide form cohesive aggregates, while start dispersing away when the byproduct of the digestion of the polysaccharide start accumulating. Dispersing cells, tend to be attracted by the polysaccharide. Expression data show that motility genes are enriched during the dispersal phase, as expected. Counterintuitively, in the same phase, genes for transporters and digestions of polysaccharide are also highly expressed.

Strengths:

The manuscript is very well written and easy to follow. The topic is interesting and timely. The genetic analyses provide a new, albeit complex, angle to the study of foraging behaviors in bacteria, adding to previous studies conducted on other species.

Weaknesses:

I find this paper very descriptive and speculative. The results of the genetic analyses are quite counterintuitive; therefore, I understand the difficulty of connecting them to the observations coming from experiments in the microfluidic device. However, they could be better placed in the literature of foraging - dispersal cycles, beyond bacteria. In addition, the interpretation of the results is sometimes confusing.

---

## [Author Response]

The following is the authors’ response to the original reviews.

Editors’ recommendations for the authorsThe reviewers recommend the following:(a) Digging deeper into the discussion of the density-dependent dispersal.(b) Clarifying the microfluidic setup.(c) Clarifying the description and interpretation of the transcriptomic evidence.(d) Toning down carbon cycle connections (some reviewers felt the evidence did not fully support the claims).

We would like to thank the editors for their thoughtful evaluation of our manuscript and their clear suggestions. We have revised the manuscript in the light of these comments, as we outline below and address in detail in the point-by-point response to the reviewers’ comments that follows.

(a) We have expanded the discussion of density-dependent dispersal and revised Figure 2C to improve clarity.

(b) We have also added further information concerning the microfluidic setup in the results section and provide an illustration of the setup in a new figure panel, Figure 1A.

(c) Addressing the reviewers’ comments on the transcriptomic analysis, we have added more information in the description and interpretation of the results.

(d) We have rephrased the text describing the role of degradation-dispersal cycles for carbon cycling to highlight it as the motivation of this study and emphasize the link to literature on foraging, without creating expectations of direct measurements of global carbon cycling.

**Public Reviews:**

**Reviewer #1 (Public Review):**
[...]Weaknesses:Much of the genetic analysis, as it stands, is quite speculative and descriptive. I found myself confused about many of the genes (e.g., quorum sensing) that pop up enriched during dispersal quite in contrast to my expectations. While the authors do mention some of this in the text as worth following up on, I think the analysis as it stands adds little insight into the behaviors studied. However, I acknowledge that it might have the potential to generate hypotheses and thus aid future studies. Further, I found the connections to the carbon cycle and marine environments in the abstract weak --- the microfluidics setup by the authors is nice, but it provides limited insight into naturalistic environments where the spatial distribution and dimensionality of resources are expected to be qualitatively different.

We thank the reviewer for their suggestions to improve our manuscript. We agree that the original manuscript would have benefitted from more detailed interpretation of the observed changes in gene expression. We have revised the manuscript to elaborate on the interpretation of the changes in expression of quorum sensing genes (see response to reviewer 1, comment 3), motility genes (see response to reviewer 1, comment 6), alginate lyase genes (see response to reviewer 1, comment 7 and reviewer 2, comment 2), and ribosomal and transporter genes (see response to reviewer 2, comment 2).

In general, we think that the gene expression study not only supports the phenotypic observations that we made in the microfluidic device, such as the increased swimming motility when exposed to digested alginate medium, but also adds further insights. Our reasoning for studying the transcriptomes in well mixed-batch cultures was the inability to study gene expression dynamics to support the phenotypic observations about differential motility and chemotaxis in our microfluidics setup. The transcriptomic data clearly show that even in well-mixed environments, growth on digested alginate instead of alginate is sufficient to increase the expression of motility and chemotaxis genes. In addition, the finding that expression of alginate lyases and metabolic genes is increased during growth on digested alginate was revealed through the analysis of transcriptomes, something which would not have been possible in the microfluidic setup. We agree with the reviewer that our analyses implicate further, perhaps unexpected, mechanisms like quorum sensing in the cellular response to breakdown products, and that this represents an interesting avenue for further studies.

Finally, we also agree with the reviewer that it would be good to be more explicit in the text that our microfluidic system cannot fully capture the complex dynamics of natural environments. Our approach does, however, allow the characterization of cellular behaviors at spatial and temporal scales that are relevant to the interactions of bacteria, and thus provides a better understanding of colonization and dispersal of marine bacteria in a manner that is not possible through in situ experiments. We have edited our manuscript to highlight this and modified our statements regarding carbon cycling towards emphasizing the role degradation-dispersal cycles in remineralization of polysaccharides (see response to reviewer 1, comment 2).

**Reviewer #2 (Public Review):**
[...]Weaknesses:The explanation of the microfluidics measurements is somewhat confusing but I think this could be easily remedied. The quantitative interpretation of the dispersal data could also be improved and I'm not clear if the data support the claim made.

We thank the reviewer for their comments and helpful suggestions. We have revised the manuscript with these suggestions in mind and believe that the manuscript is improved by a more detailed explanation of the microfluidic setup. We have added more information in the text (detailed in response to reviewer 2, comments 1 and 2) and have added a depiction of the microfluidic setup (Fig. 1A). We have also modified the presentation and discussion of the dispersal data (Fig. 2C), as described in detail below in response to reviewer 2, comment 4, and argue that they clearly show density-dependent dispersal. We believe that this modification of how the results are presented provides a more convincing case for our main conclusion, namely that the presence of degradation products controls bacterial dispersal in a density-dependent manner.

**Reviewer #3 (Public Review):**
[...]Weaknesses:I find this paper very descriptive and speculative. The results of the genetic analyses are quite counterintuitive; therefore, I understand the difficulty of connecting them to the observations coming from experiments in the microfluidic device. However, they could be better placed in the literature of foraging - dispersal cycles, beyond bacteria. In addition, the interpretation of the results is sometimes confusing.

We thank the reviewer for their suggestions to improve the manuscript. We have edited the manuscript to interpret the results of this study more clearly, in particular with regard to the fact that breakdown products of alginate cause cell dispersal (see response to reviewer 2, comment 1), gene expression changes of ribosomal proteins and transporters (see response to reviewer 2, comment 2), as well as genes relating to alginate catabolism (see response to reviewer 2, comment 3).

To provide more context for the interpretation of our results we now also embed our findings in more detail in the previous work on foraging strategies and dispersal tradeoffs.

**Recommendations For The Authors:**

**Reviewer #1 (Recommendations For The Authors):**
(1) The authors should clarify in more detail what they mean by density dependence in Figure 2. Usually density dependence refers to a per capita dependence, but here it seems that the per capita rate of dispersal might be roughly independent of density (Figure 2c; if you double the number of cells it doubles the number of cells leaving). Rather it seems the dispersal is such that the density of remaining cells falls below a threshold (~300 cells).

We thank the reviewer for raising this important point. To analyze the data more explicitly in terms of per capita dependence and so make the density dependence in the dispersal from the microfluidic chambers more clear, we have modified Figure 2C and edited the text.

In the modified Figure 2C, we computed the fraction of dispersed cells for each chamber (i.e the change in cell number divided by the cell number at the time of the nutrient switch). This quantity directly reveals the per-capita dependence, as mentioned by reviewer 1, and is now represented on the y-axis of Figure 2C instead of the absolute change in cell number.

These data demonstrate that the fraction of dispersed cells increases with increasing numbers of cells present in the chamber at the time of switching, with more highly populated chambers showing a higher fraction of dispersed cells. These findings indicate that there is a strong density dependence in the dispersal process.

As pointed out by reviewer 1, another interesting aspect of the data is the transition at low cell number. The fraction of dispersed cells is negative in the case of the chamber with approximately 70 cells, consistent with no dispersal at this low density, and a moderate density increase as a function of continued growth.

In addition to the new analysis presented in Figure 2C, we have modified the paragraph that discusses this result as follows (line 208):

“We indeed found that the nutrient switch caused a few or no cells to disperse from small cell groups (Fig. 2B), whereas a large fraction of cells from large cell groups dispersed (Fig. 2C). In fact, the e fraction of cells that dispersed upon imposition of the nutrient switch showed a strong positive relationship with the number of cells present, meaning that cells in chambers with many cells were more likely to disperse than cells in chambers with fewer cells (Fig. 2C).”

(2) The authors should tone down their claims about the carbon cycle in the abstract. I do not believe the results as they stand could be used to understand degradation-dispersal cycles in marine environments relevant to the carbon cycle, since these behaviors have been studied in microfluidic environments which in my understanding are quite different. As such, statements such as "degradation-dispersal cycles are an integral part in the global carbon cycle, we know little about how cells alternate between degradation and motility" and "Overall, our findings reveal the cellular mechanisms underlying bacterial degradation-dispersal cycles that drive remineralization in natural environments" are overstated in the abstract.

We appreciate the reviewer’s comments regarding the connections of our work with the carbon cycle. We have now rephrased these statements in our manuscript to describe a potential connection between our work and the marine carbon cycle. The colonization of polysaccharides particles by bacteria and subsequent degradation has been widely acknowledged to play a significant role in controlling the carbon flow in marine ecosystems. (Fenchel, 2002; Preheim et al., 2011; Yawata et al., 2014, 2020). We still refer to carbon flow in the revised manuscript, though cautiously, as microbial remineralization of biomass, which is recognized as an important factor in the marine biological carbon pump (e.g., Chisholm, 2000; Jiao et al., 2024). As stated in the previous version of the manuscript, the main motivation of our work was to study the growth behaviors of marine heterotrophic bacteria during polysaccharide degradation, especially to understand when bacteria depart already colonized and degraded particles and find novel patches to grow and degrade, a process that is poorly understood. Therefore, it is conceivable that degradation-dispersal cycles do play a role in the flow of carbon in marine ecosystems. However, we acknowledge that the carbon cycle is influenced by a multitude of biological and chemical processes, and the bacterial degradation-dispersal cycle might not be the sole mechanism at play.

We also appreciate the reviewer’s comments highlighting that the complexity of natural environments is not fully captured in our microfluidics system. However, our microfluidics setup does allow us to quantify responses and behaviors of microbial groups at high spatial and temporal resolution, especially in the context of environmental fluctuations. Microbes in nature interact at small spatial scales and have to respond to changes in the environment, and the microfluidics setup enables the quantification of these responses. Moreover, dispersal of the bacterium V. cyclitrophicus that we use in our study, has been previously observed even during growth on particulate alginate (Alcolombri et al., 2021), but the cues and regulation controlling dispersal behaviors have been unclear. Microfluidic experiments have now allowed us to study this process in a highly quantitative manner, and align well with observations from experiments from more nature-like settings. These quantitative experiments on bacterial strains isolated from marine particles are expected to constrain quantitative models of carbon degradation in the ocean (Nguyen et al., 2022).

We have now adjusted our statements throughout our manuscript to reflect the knowledge gaps in understanding the triggers of degradation-dispersal cycles and their links with carbon flow in marine ecosystems. The revised manuscript, especially, contains the following statements (line 47 and line 60):

“Even though many studies indicate that these degradation-dispersal cycles contribute to the carbon flow in marine systems, we know little about how cells alternate between polysaccharide degradation and motility, and which environmental factors trigger this behavioral switch.”

“Overall, our findings reveal cellular mechanisms that might also underlie bacterial degradation-dispersal cycles, which influence the remineralization of biomass in marine environments.”

(3) The authors should clarify why they think quorum-sensing genes are increased in expression on digested alginate. The authors currently mention that QS could be used to trigger dispersal, but given the timescales of dispersal in Figure 2 (~half an hour), I find it hard to believe that these genes are expressed and have the suggested effect on those timescales. As such I would have expected the other way round - for QS genes to be expressed highly during alginate growth, so that density could be sensed and responded to. Please clarify.

We have now clarified this point in the revised manuscript. While the triggering of dispersal by quorum-sensing genes may indeed appear counterintuitive, and the response is rapid (we see dispersal of cells within 30-40 minutes), both observations are in line with previous studies in another model organism *Vibrio cholerae*. The dispersal time is similar to the dispersal time of *V. cholerae* cells from biofilms, as described by Singh and colleagues, (Figure 1E of Ref. Singh et al., 2017). In that case, induction of the quorum sensing dispersal regulator HapR was observed during biofilm dispersal within one hour after switch of condition (Fig. 2, middle panel of Ref. Singh et al., 2017). Even though the specific quorum sensing signaling molecules are probably different in our strain (there is no annotated homolog of the hapR gene in V. cyclitrophicus), we observed that the full set of quorum sensing genes was enriched in cells growing on digested alginate (as reported in line 314 and Fig. 4A).

We have added this information in the manuscript (line 317):

“The set of quorum sensing genes was also positively enriched in cells growing on digested alginate (Fig. 4A and S4F, Table S13). This role in dispersal is in agreement with a previous study that showed induction of the quorum sensing master regulator in *V. cholerae* cells during dispersal from biofilms on a similar time scale as here (less than an hour) [28].”

**Reviewer #2 (Recommendations For The Authors):**
(1) Around line 144 - I don't really understand how you flow alginate through the microfluidic platform. It seems if the particles are transiently going through the microfluidic chamber then the flow rate and hence residence time of the alginate particles will matter a lot by controlling the time the cells have to colonize and excrete enzymes for alginate breakdown. Or perhaps the alginate is not particulate but is instead a large but soluble polymer? I think maybe a schematic of the microfluidic device would help -- there is an implicit assumption that we are familiar with the Dal Co et al device, but I don't recall its details and maybe a graphic added to Figure 1 would help.a. In reviewing the Dal Co paper I see that cells are trapped and the medium flows through channels and the plane where the cells are held. I am still a little confused about the size of the polymeric alginate -- large scale (>1um) particles or very small polymers?

We have now provided a detailed description of our microfluidic experimental system. At the start of the experiments, cells are in fact not trapped within the microfluidic device, but grow and can move freely within a chamber designed with dimensions (sub-micron heights) so that growth occurs only as a monolayer. Cells were exposed to nutrients, either alginate or alginate digestion products, both in soluble form (not particles). These compounds were flowed into the device through a main channel, but entered the flowfree growth chambers by diffusion. To make these aspects of our experiments clearer, we have added further information on this in the Materials & Methods section (line 556), added this information in the abstract (line 51), and in the results (line123).

To make our microfluidic setup clearer, we have followed this advice and added a schematic as Figure 1A and have added more information on the setup to the main text (line 153):

“In brief, the microfluidic chips are made of an inert polymer (polydimethylsiloxane) bound to a glass coverslip. The PDMS layer contains flow channels through which the culture medium is pumped continuously. Each channel is connected to several growth chambers that are laterally positioned. The dimensions of these growth chambers (height: 0.85 µm, length: 60 µm, width: 90-120 µm) allow cells to freely move and grow as monolayers. The culture medium, containing either alginate or digested alginate in their soluble form, is constantly pumped through the flow channel and enters the growth chambers primarily through diffusion [15,16,4,17,8]. Therefore, the number of cells and their positioning within microfluidic chambers is determined by the cellular growth rate as well as by cell movement4. This setup combined with time-lapse microscopy allowed us to follow the development of cell communities over time.”

(2) What makes this confusing is the difference between Figure 1C and Figure S2A -- the authors state that the difference in Figure 1C is due to dispersal, but is there flow through the microfluidic device? So what role does that flow through the device have in dispersal? Is the adhesion of the cell groups driven at all by a physical interaction with high molecular weight polymers in the microfluidic devices or is this purely a biological effect? Could this also be explained by different real concentrations of nutrients in the two cases?

We realize from this comment that the role of flow of the medium in the microfluidic setup was not clearly addressed in our manuscript. In fact, cells were not exposed to flow, and nutrients were provided to the growth chambers by diffusion. We have added a clearer explanation of this point on line 158:

“The culture medium, containing either alginate or digested alginate in their soluble form, is constantly pumped through the flow channel and enters the growth chambers primarily through diffusion [15,16,4,17,8]. Therefore, the number of cells and their positioning within microfluidic chambers is determined by the cellular growth rate as well as by cell movement4.“

One purely physical effect that we anticipate is that a high viscosity of the medium could immobilize cells. To address this point, we measured the viscosity of both alginate and digested alginate and conclude that the increase in viscosity is not strong enough to immobilize cells. We added a statement in the text (line 170)

“To test the role of increased viscosity of polymeric alginate in causing the increased aggregation of cells, we measured the viscosity of 0.1% (w/v) alginate or digested alginate dissolved in TR media. For alginate, the viscosity was 1.03±0.01 mPa·s (mean and standard deviation of three technical replicates) whereas the viscosity of digested alginate in TR media was found to be 0.74±0.01 mPa·s. Both these values are relatively close to the viscosity of water at this temperature (0.89 mPa·s18) and, while they may affect swimming behavior [19], they are insufficient to physically restrain cell movement [20].”

as well as a section in the Materials and Methods (line 594):

“Viscosity of the alginate and digested alginate solution

We measured the viscosity of alginate solutions using shear rheology measurements. We use a 40 mm cone-plate geometry (4° cone) in a Netzsch Kinexus Pro+ rheometer. 1200 uL of sample was placed on the bottom plate, the gap was set at 150 um and the sample trimmed. We used a solvent trap to avoid sample evaporation during measurement. The temperature was set to 25°C using a Peltier element. We measure the dynamic viscosity over a range of shear rates = 0.1 – 100 s-1. We report the viscosity of each solution as the average viscosity measured over the shear rates 10 – 100 s-1, where the shear-dependence of the viscosity was low.

We measured the viscosity of 0.1% (w/V) alginate dissolved in TR media, which was 1.03 +/- 0.01 mPa·s (reporting the mean and standard deviation of three technical replicates.). The viscosity of 0.1% digested alginate in TR media was found to be 0.74+/-0.01 mPa·s. This means that the viscosity of alginate in our microfluidic experiments is 36% higher than of digested alginate, but the viscosities are close to those expected of water (0.89 mPa·s at 25 degree Celsius according to Berstad and colleagues [18]).”

While our microfluidic setup allows us to track the position and movement of cells in a spatially structured setting, these observations do not allow us to distinguish directly whether the differences in dispersal are a result of purely physical effects of polymers on cells or are a result of them triggering a biological response in cells that causes them to become sessile. It is known that bacterial appendages like pili interact with polysaccharide residues (Li et al., 2003). Therefore, it is quite plausible that cross-linking by polysaccharides can contribute growth behaviors on alginate. However, our analysis of gene expression demonstrates that flagellum-driven motility is decreased in the presence of alginate compared to digested alginate, alongside other major changes in gene expression. In addition, our measures of dispersal show that dispersal of cells when exposed to digested alginate is density dependent. Both observations suggest that the patterns in dispersal are governed by decision-making processes by cells resulting in changes in cell motility, rather than being a product of purely physical interactions with the polymer.

The finding that viscosities of both alginate and digested alginate are similar to that of water, suggests that diffusion of nutrients in the growth chambers should be similar. Therefore, we think that the differences in real concentrations of nutrients is likely not contributing to the observed differences in behavior.

(3) Why is Figure S1 arbitrary units? Does this have to do with the calibration of LC-MS? It would be better, it seems, to know the concentrations in real units of the monomer at least.

We agree with the reviewer that it would have been better to have absolute concentrations for these compounds. However, to calibrate the mass spectrometer signals (ion counts) to absolute concentrations for the different alginate compounds, we would need an analytical standard of known concentration. We are not aware of such a standard and thus report only relative concentrations. We agree that the y-axis label of Figure S1 should not contain ‘arbitrary’ units, as it shows a ratio (of measurements in the same arbitrary units). We have edited the labels of Figure S1 accordingly and the figure legend in line 26 of the Supplemental Material (“Relative concentrations…”).

(4) Line 188 - density-dependent dispersal. The claim here is that "cells in chambers with many cells were more likely to disperse than cells in chambers with less cells." (my emphasis). Looking at the data in Figure 2C it appears that about 40% of the cells disperse irrespective of the density, before the switch to digested alginate. So it would seem that there is not a higher likelihood of dispersal at higher cell densities. For the very highest cell density, it does appear that this fraction is larger, but I'd be concerned about making this claim from what I understand to be a single experiment. To support the claim made should the authors plot Change in Cell number/Starting Cell number on the y-axis of Fig. 2C to show that the fraction is increasing? It would seem some additional data at higher starting cell densities would help support this claim more strongly.

We thank the reviewer for this comment, which is in line with a remark made by reviewer 1 in their comment 1. In response to these two comments (and as described above), we have edited Figure 2C and now have plotted the change in cell number relative to starting cell number at the y axis to directly show the density dependence. We observe a positive (approximately linear) relationship between the fraction of dispersed cells with the number of cells present in the chamber at the time of switching. This indicates that there is a density dependence in the dispersal process, with highly populated chambers showing a higher fraction of dispersed cells.

In addition to the change in Figure 2C, we have modified the paragraph around line 208: “We indeed found that the nutrient switch caused a few or no cells to disperse from small cell groups (Fig. 2B), whereas a large fraction of cells from large cell groups dispersed (Fig. 2C). In fact, the e fraction of cells that dispersed upon imposition of the nutrient switch showed a strong positive relationship with the number of cells present, meaning that cells in chambers with many cells were more likely to disperse than cells in chambers with fewer cells (Fig. 2C).”

The highest cell number at the start of the switch that we include is about 800 cells. The maximum number of cells that can fit into a chamber are ca. 1000 cells. Thus, 800 resident cells are close to the maximal density.

(5) A comment -- I find the result of significant chemotaxis towards alginate but not the monomers of alginate to be quite surprising. The ecological relevance of this (line 219) seems like an important result that is worth expanding on a bit at least in the discussion. For now, my question is whether the authors know of any mechanism by which chemotaxis receptors could respond to alginate but not the monomer. How can a receptor distinguish between the two?

We agree that this result is surprising, given that oligomers can be more easily transported into the periplasm where sensing takes place, and they also provide an easier accessible nutrient source. Indeed, in case of the insoluble polymer chitin it has been shown that chemotaxis towards chitin is mediated by chitin oligomers (Bassler et al., 1991), which was suggested as a general motif to locate polysaccharide nutrient sources (Keegstra et al., 2022). However, a recent study has changed this perspective by showing widespread chemotaxis of marine bacteria towards the glucose-based marine polysaccharide laminarin, but not towards laminarin oligomers or glucose (Clerc et al., 2023). Together with our results on chemotaxis towards alginate (but not significantly toward alginate oligomers) this suggests that chemotaxis towards soluble polysaccharides can be mediated by direct sensing of the polysaccharide molecules.

As recommended, we expanded the discussion of the ecological relevance and also added more information on possible mechanisms of selective sensing of alginate and its breakdown products (around line 479).:

“Direct chemotaxis towards polysaccharides may facilitate the search for new polysaccharide sources after dispersal. We found that the presence of degradation products not only induces cell dispersal but also increases the expression of chemotaxis genes. Interestingly, we found that V. cyclitrophicus ZF270 cells show chemotaxis towards polymeric alginate but not digested alginate. This contrasts with previous findings for bacterial strains degrading the insoluble marine polysaccharide chitin, where chemotaxis was strongest towards chitin oligomers53, suggesting that oligomers may act as an environmental cue for polysaccharide nutrient sources55. However, recent work has shown that certain marine bacteria are attracted to the marine polysaccharide laminarin, and not laminarin oligomers56. Together with our results, this indicates that chemotaxis towards soluble polysaccharides may be mediated by the polysaccharide molecules themselves. The mechanism of this behavior is yet to be identified, but could be mediated by polysaccharide-binding proteins as have been found in Sphingomonas sp. A1 facilitating chemotaxis towards pectin57. Direct polysaccharide sensing adds complexity to chemosensing as polysaccharides cannot freely diffuse into the periplasm, which can lead to a trade-off between chemosensing and uptake58. Furthermore, most polysaccharides are not immediately metabolically accessible as they require degradation. But direct polysaccharide sensing can also provide certain benefits compared to using oligomers as sensory cues. First, it could enable bacterial strains to preferably navigate to polysaccharide nutrients sources that are relatively uncolonized and hence show little degradation activity. Second, strong chemotaxis towards degradation products could hinder a timely dispersal process as the dispersal then requires cells to travel against a strong attractant gradient formed by the degradation products. Overall, this strategy allows cells to alternate between degradation and dispersal to acquire carbon and energy in a heterogeneous world with nutrient hotspots [44,59–61].”

(6) Comment on lines 287-8 -- that the "positive enrichment of the gene set containing bacterial motility proteins matched the increase in motile cells that we observe in Fig 3E." I'm confused about what is meant by the word "matched" here. Is the implication that there is some quantitative correspondence between increased motility in Figure 3 and the change in expression in Figure 4? Or is the statement a qualitative one -- that motility genes are upregulated in the presence of digested alginate? Table S12 didn't help me answer this question.

We thank the reviewer for their helpful comment. Our original statement was a qualitative one - observing that gene expression enrichment in genes associated with bacterial motility aligned with our expectations based on the previous observation of an increase in motile cells. We have now changed the wording to highlight the qualitative nature of this statement (line 315):

“The positive enrichment of the gene set containing bacterial motility proteins aligned with our expectations based on the increase in motile cells that we observed in Figure 3E (Fig. 4A, Table S12).”

(7) Line 326 - what is the explanation for the production of public enzymes in the presence of digest? How does this square with the previous narrative about cells growing on alginate digest expressing motility genes and chemotaxing towards alginate? It seems like the story is a bit tenuous here in the sense that digested alginates stimulate both motility - which is hypothesized to drive the discovery of new alginate particles - and lyase enzymes which are used to degrade alginate. So do the high motility cells that are chemotaxing towards alginate also express lyases en route? I'm of the opinion that constructing narratives like these in the absence of a more quantitative understanding of the colonization and degradation dynamics of alginate particles presents a major challenge and may be asking more of the data than the data can provide.a. I noted later that this is addressed later around lines 393 in the Discussion section.

Indeed, the notion that the presence of breakdown products triggers motility and also increases the expression of alginate lyases and other metabolic genes for alginate catabolism seems counterintuitive. We have now expanded our discussion of these results to contextualize these findings (around line 443):

"One reason for this observation may be that cells primarily rely on intracellular monosaccharide levels to trigger the upregulation of genes associated with polysaccharide degradation and catabolism, as has previously been observed for *E. coli* across various carbon sources [50,51]. In fact, the majority of carbon sources are sensed by prokaryotes through one‑component sensors inside the cell50. In the one‑component internal sensing scheme, the enzymes and transporters for the use of various carbon sources are expressed at basal levels, which leads to an increase in pathway intermediates upon nutrient availability. The pathway intermediates are sensed by an internal sensor, usually a transcription factor, and lead to the upregulation of transporter and enzyme expression [50,51]. This results in a positive feedback loop, which enables small changes in substrate abundance to trigger large transcriptional responses [50,52]. Thus, the presence of alginate breakdown products may likely result in increased expression of all components of the alginate degradation pathway, including the expression of degrading enzymes. As the gene expression analysis was performed on well-mixed cultures in culture medium containing alginate breakdown products, we therefore expect a strong stimulation of alginate catabolism. In a natural scenario, where cells disperse from a polysaccharide hotspot before its exhaustion, the expression of alginate catabolism genes may likely decrease again once the local concentration of breakdown products decreases. However, continued production of alginate lyases could also provide an advantage when encountering a new alginate source and continued production of alginate lyases may thus help cells to prepare for likely future environments. Further investigations of bacterial enzyme secretion in changing nutrient environments and at relevant spatial scales are required to improve our understanding of the regulation of enzyme secretion along nutrient gradients."

(8) I like Figure 6, and I think this hypothesis is a good result from this paper, but I think it would be important to emphasize this as a proposal that needs further quantitative analysis to be supported.

We have now edited the manuscript to make this point more clear. While both degradation and dispersal are well-appreciated parts of microbial ecology, the transitions and underlying mechanisms are unclear. We have edited the discussion to improve the clarity (line 419):

“This cycle of biomass degradation and dispersal has long been discussed in the context of foraging e.g., [44,45,13,46,47], but the cellular mechanisms that drive the cell dispersal remain unclear.”

Also, we have updated Figure 6 to indicate more clearly which new findings this work proposes (now bold font) and which previous findings that were made in different bacterial taxa and carbon sources that aligns with our work (now light font). We edited the figure legend accordingly (line 503):

"By integrating our results with previous studies on cooperative growth on the same system, as well as results on dispersal cycles in other systems, we highlight where the specific results of this work add to this framework (bold font)."

Minor comments(1) Is there any growth on the enzyme used for alginate digestion? E.g. is the enzyme used to digest the alginate at sufficiently high concentrations that cells could utilize it for a carbon/nitrogen source?

We thank the reviewer for raising this point. We added the following paragraph as Supplemental Text to address it (line 179):

“Protein amount of the alginate lyases added to create digested alginate

Based on the following calculation, we conclude that the amount of protein added to the growth medium by the addition of alginate lyases is so small that we consider it negligible. In our experiment we used 1 unit/ml of alginate lyases in a 4.5 ml solution to digest the alginate. As the commercially purchased alginate lyases are 10,000 units/g, our 4.5 ml solution contains 0.45 mg of alginate lyase protein. The digested alginate solution diluted 45x when added to culture medium. This means that we added 0.18 µg alginate lyase protein to 1 ml of culture medium.

As a comparison, for 1ml of alginate medium, 1000µg of alginate is added or for 1 ml of Lysogeny broth (LB) culture medium, 3,500 µg of LB are added. Thus, the amount of alginate lyase protein that we added is ca. 5000 - 20,000 times smaller than the amount of alginate or LB that one would add to support cell growth. Therefore, we expect the growth that the digestion of the added alginate lyases would allow to be negligible.”

(2) The lines in Figure 2B are very hard to see.

We have addressed this comment by using thicker lines in Figure 2B.

(3) The black background and images in Figure 3A and B are hard to see as well.

We have now replaced Figure 3A and B, now using a white background.

(4) Typo at the beginning of line 251?

Unfortunately we failed to find the typo referred to. We are happy to address it if it still exists in the revised manuscript.

**Reviewer #3 (Recommendations For The Authors):**
(1) I think there is not enough experimental evidence to conclude that the underlying cause of increased motility is the accumulation of digested alginate products. To conclusively show that this is the cause and not just some signal linked to cell density, perhaps the experiment should be repeated with a different carbon source.

We thank the reviewer for their comment, which made us realize that we did not make the nature of the dispersal cue clear. The gene expression data was obtained from batch cultures and measured at the same approximate bacterial densities in batch, which indeed shows that the digested alginate is a sufficient signal for an increase in motility gene expression. This agrees very well with our observation that cells growing on digested alginate in microfluidic chambers have an increased fraction of motile cells in comparison with cells exposed to alginate (Fig 3E). However, we did not mean to suggest that the observed dispersal by bacterial motility is not influenced by cell density, in fact, we see that dispersal (and hence the increase in cell motility) in microfluidic chambers that are switched from polymeric to digested alginate depends on the bacterial density in the chamber, with higher bacterial densities showing increased dispersal. This shows that the presence of alginate oligomers does trigger dispersal through motility, but this signal affects bacterial groups in a cell density dependent manner.

Similar observations have been made in Caulobacter crescentus, which was found to form cell groups on the polymer xylan while cells disperse when the corresponding monomer xylose becomes available (D’Souza et al., 2021). We reference the additional work in lines 179 and 230. Taken together, these observations indicate a more general phenomenon in dispersal from polysaccharide substrates.

(2) About the expression data:• Ribosomal proteins and ABC transporters are enriched in cells grown on digested alginate and the authors discuss that this explains the difference in max growth rate between alginate and digested alginate. However, in Figure S2E the authors report no statistical difference between growth rates.

We have now edited the manuscript to clarify this point. We found that cells grown on degradation products reached their maximal growth rate around 7.5 hours earlier (Fig. S2D) and showed increased expression of ribosomal biosynthesis and ABC transporters in late-exponential phase (Fig. 4A). We consider this shorter lag time as a sign of a different growth state and therefore a possible reason for the difference in ribosomal protein expression.

As the reviewer correctly points out, the maximum growth rates that were computed from the two growth curves were not significantly different (Fig. S2E). However, for our gene expression analysis, we harvested the transcriptome of cells that reached OD 0.39-0.41 (mid- to late-exponential phase). At this time point, the cell cultures may have differed in their momentary growth rate.

We edited the manuscript to make this clearer (line 287):

“Both observations likely relate to the different growth dynamics of V. cyclitrophicus ZF270 on digested alginate compared to alginate (Fig. S2A), where cells in digested alginate medium reached their maximal growth rate 7.5 hours earlier and thus showed a shorter lag time (Fig. S2D). As a consequence, the growth rate at the time of RNA extraction (mid-to-late exponential phase) may have differed, even though the maximum growth rate of cells grown in alginate medium and digested alginate medium were not found to be significantly different (Fig. S2E).”

• The increased expression of transporters for lyases in cells grown on digested alginate (lines 273-274 and 325-328) is very confusing and the explanation provided in lines 412-420 is not very convincing. My two cents on this: Expression of more enzymes and induction of motility might be a strategy to be prepared for more likely future environments (after dispersal, alginate is the most likely carbon source they will find). This would be in line with observed increased chemotaxis towards the polymer rather than the monomer (Similar to *C. elegans*).

This comment is in line with reviewer 2, comment 7. In response to these two comments (and as described above), we expanded our discussion of these results to contextualize these findings (around line 443):

“One reason for this observation may be that cells primarily rely on intracellular monosaccharide levels to trigger the upregulation of genes associated with polysaccharide degradation and catabolism, as has previously been observed for *E. coli* across various carbon sources [50,51]. In fact, the majority of carbon sources are sensed by prokaryotes through one‑component sensors inside the cell [50]. In the one‑component internal sensing scheme, the enzymes and transporters for the use of various carbon sources are expressed at basal levels, which leads to an increase in pathway intermediates upon nutrient availability. The pathway intermediates are sensed by an internal sensor, usually a transcription factor, and lead to the upregulation of transporter and enzyme expression [50,51]. This results in a positive feedback loop, which enables small changes in substrate abundance to trigger large transcriptional responses [50,52]. Thus, the presence of alginate breakdown products may likely result in increased expression of all components of the alginate degradation pathway, including the expression of degrading enzymes. As the gene expression analysis was performed on well-mixed cultures in culture medium containing alginate breakdown products, we therefore expect a strong stimulation of alginate catabolism. In a natural scenario, where cells disperse from a polysaccharide hotspot before its exhaustion, the expression of alginate catabolism genes may likely decrease again once the local concentration of breakdown products decreases. However, continued production of alginate lyases could also provide an advantage when encountering a new alginate source and continued production of alginate lyases may thus help cells to prepare for likely future environments. Further investigations of bacterial enzyme secretion in changing nutrient environments and at relevant spatial scales are required to improve our understanding of the regulation of enzyme secretion along nutrient gradients.”

Additionally, we agree with the intriguing comment that continued expression of alginate lyases may also prepare cells for likely future environments. Further studies that aim to answer whether marine bacteria are primed by their growth on one carbon source towards faster re-initiation of degradation on a new particle will be an interesting research question. We now address this point in our manuscript (line 458):

“However, continued production of alginate lyases could also provide an advantage when encountering a new alginate source and continued production of alginate lyases may thus help cells to prepare for likely future environments. Further investigations of bacterial enzyme secretion in changing nutrient environments and at relevant spatial scales are required to improve our understanding of the regulation of enzyme secretion along nutrient gradients.“

(3) The yield reached by Vibrio on alginate is significantly higher than the yield in digested alginate, not similar, as stated in lines 133-134. Only cell counts are similar. Perhaps the author can correct this statement and speculate on the reason leading to this discrepancy: perhaps cells tend to aggregate in alginate despite the fact that these are well-mixed cultures.

We have edited the description of the OD measurements accordingly and agree with the reviewer that aggregation is indeed a possible reason for the discrepancy (line 141):

“We also observed that the optical density at stationary phase was higher when cells were grown on alginate (Fig. S2B and C). However, colony counts did not show a significant difference in cell numbers (Fig. S3), suggesting that the increased optical density may stem from aggregation of cells in the alginate medium, as observed for other Vibrio species [7].”

(4) I suggest toning down the importance of the results presented in this study for understanding global carbon cycling. There is a link but at present it is too much emphasized.

We have edited our statements regarding the carbon cycle. In the revised manuscript we stress the lack of direct quantifications of carbon cycling. . We still refer to carbon flow in the revised manuscript, as we would argue that microbial remineralization of biomass is recognized as an important factor in the marine biological carbon pump (e.g., Chisholm, 2000) and research on marine bacterial foraging investigates how bacterial cells manage to find and utilize this biomass.

Our revised manuscript contains the following modified statements (line 47 and line 60): “Even though many studies indicate that these degradation-dispersal cycles contribute to the carbon flow in marine systems, we know little about how cells alternate between polysaccharide degradation and motility, and which environmental factors trigger this behavioral switch.”

“Overall, our findings reveal cellular mechanisms that might also underlie bacterial degradation-dispersal cycles, which influence the remineralization of biomass in marine environments.”

**References**

Alcolombri, U., Peaudecerf, F. J., Fernandez, V. I., Behrendt, L., Lee, K. S., & Stocker, R. (2021). Sinking enhances the degradation of organic particles by marine bacteria. Nature Geoscience, 14(10), 775–780. https://doi.org/10.1038/s41561-021-00817-xBassler, B. L., Gibbons, P. J., Yu, C., & Roseman, S. (1991). Chitin utilization by marine bacteria. Chemotaxis to chitin oligosaccharides by Vibrio furnissii. Journal of Biological Chemistry, 266(36), 24268–24275. https://doi.org/10.1016/S0021-9258(18)54224-1Chisholm, S. W. (2000). Stirring times in the Southern Ocean. Nature, 407(6805), 685–686. https://doi.org/10.1038/35037696Chubukov, V., Gerosa, L., Kochanowski, K., & Sauer, U. (2014). Coordination of microbial metabolism. Nature Reviews. Microbiology, 12(5), 327–340. https://doi.org/10.1038/nrmicro3238Clerc, E. E., Raina, J.-B., Keegstra, J. M., Landry, Z., Pontrelli, S., Alcolombri, U., Lambert, B. S., Anelli, V., Vincent, F., Masdeu-Navarro, M., Sichert, A., De Schaetzen, F., Sauer, U., Simó, R., Hehemann, J.-H., Vardi, A., Seymour, J. R., & Stocker, R. (2023). Strong chemotaxis by marine bacteria towards polysaccharides is enhanced by the abundant organosulfur compound DMSP. Nature Communications, 14(1), 8080. https://doi.org/10.1038/s41467-023-43143zDal Co, A., van Vliet, S., Kiviet, D. J., Schlegel, S., & Ackermann, M. (2020). Shortrange interactions govern the dynamics and functions of microbial communities. Nature Ecology and Evolution, 4(3), 366–375. https://doi.org/10.1038/s41559-019-1080-2D’Souza, G., Ebrahimi, A., Stubbusch, A., Daniels, M., Keegstra, J., Stocker, R., Cordero, O., & Ackermann, M. (2023). Cell aggregation is associated with enzyme secretion strategies in marine polysaccharide-degrading bacteria. The ISME Journal. https://doi.org/10.1038/s41396-023-01385-1D’Souza, G. G., Povolo, V. R., Keegstra, J. M., Stocker, R., & Ackermann, M. (2021). Nutrient complexity triggers transitions between solitary and colonial growth in bacterial populations. The ISME Journal, 15(9), 2614–2626. https://doi.org/10.1038/s41396-021-00953-7D’Souza, G., Schwartzman, J., Keegstra, J., Schreier, J. E., Daniels, M., Cordero, O. X., Stocker, R., & Ackermann, M. (2023). Interspecies interactions determine growth dynamics of biopolymer-degrading populations in microbial communities. Proceedings of the National Academy of Sciences of the United States of America, 120(44), e2305198120. https://doi.org/10.1073/pnas.2305198120Fenchel, T. (2002). Microbial Behavior in a Heterogeneous World. Science, 296(5570), 1068–1071. https://doi.org/10.1126/science.1070118Jiao, N., Luo, T., Chen, Q., Zhao, Z., Xiao, X., Liu, J., Jian, Z., Xie, S., Thomas, H., Herndl, G. J., Benner, R., Gonsior, M., Chen, F., Cai, W.-J., & Robinson, C. (2024). The microbial carbon pump and climate change. Nature Reviews Microbiology. https://doi.org/10.1038/s41579-024-01018-0Keegstra, J. M., Carrara, F., & Stocker, R. (2022). The ecological roles of bacterial chemotaxis. Nature Reviews Microbiology, 20(8), 491–504. https://doi.org/10.1038/s41579-022-00709-wKonishi, H., Hio, M., Kobayashi, M., Takase, R., & Hashimoto, W. (2020). Bacterial chemotaxis towards polysaccharide pectin by pectin-binding protein. Scientific Reports, 10(1), 3977. https://doi.org/10.1038/s41598-020-60274-1Li, Y., Sun, H., Ma, X., Lu, A., Lux, R., Zusman, D., & Shi, W. (2003). Extracellular polysaccharides mediate pilus retraction during social motility of Myxococcus xanthus. Proceedings of the National Academy of Sciences, 100(9), 5443–5448. https://doi.org/10.1073/pnas.0836639100Martínez-Antonio, A., Janga, S. C., Salgado, H., & Collado-Vides, J. (2006). Internal sensing machinery directs the activity of the regulatory network in *Escherichia coli*. Trends in Microbiology, 14(1), 22–27. https://doi.org/10.1016/j.tim.2005.11.002McDougald, D., Rice, S. A., Barraud, N., Steinberg, P. D., & Kjelleberg, S. (2012). Should we stay or should we go: Mechanisms and ecological consequences for biofilm dispersal. Nature Reviews Microbiology, 10(1), 39–50. https://doi.org/10.1038/nrmicro2695Nguyen, T. T. H., Zakem, E. J., Ebrahimi, A., Schwartzman, J., Caglar, T., Amarnath, K., Alcolombri, U., Peaudecerf, F. J., Hwa, T., Stocker, R., Cordero, O. X., & Levine, N. M. (2022). Microbes contribute to setting the ocean carbon flux by altering the fate of sinking particulates. Nature Communications, 13(1), 1657. https://doi.org/10.1038/s41467-022-29297-2Norris, N., Alcolombri, U., Keegstra, J. M., Yawata, Y., Menolascina, F., Frazzoli, E., Levine, N. M., Fernandez, V. I., & Stocker, R. (2022). Bacterial chemotaxis to saccharides is governed by a trade-off between sensing and uptake. Biophysical Journal, 121(11), 2046–2059. https://doi.org/10.1016/j.bpj.2022.05.003Povolo, V. R., D’Souza, G. G., Kaczmarczyk, A., Stubbusch, A. K., Jenal, U., & Ackermann, M. (2022). Extracellular appendages govern spatial dynamics and growth of *Caulobacter crescentus* on a prevalent biopolymer. bioRxiv, 2022.06.13.495907. https://doi.org/10.1101/2022.06.13.495907Preheim, S. P., Boucher, Y., Wildschutte, H., David, L. A., Veneziano, D., Alm, E. J., & Polz, M. F. (2011). Metapopulation structure of Vibrionaceae among coastal marine invertebrates. Environmental Microbiology, 13(1), 265–275. https://doi.org/10.1111/j.1462-2920.2010.02328.xSchwartzman, J. A., Ebrahimi, A., Chadwick, G., Sato, Y., Orphan, V., & Cordero, O. X. (2021). Bacterial growth in multicellular aggregates leads to the emergence of complex lifecycles. bioRxiv, 2021.11.01.466752. https://doi.org/10.1101/2021.11.01.466752Singh, P. K., Bartalomej, S., Hartmann, R., Jeckel, H., Vidakovic, L., Nadell, C. D., & Drescher, K. (2017). *Vibrio cholerae* Combines Individual and Collective Sensing to Trigger Biofilm Dispersal. Current Biology, 27(21), 3359-3366.e7. https://doi.org/10.1016/j.cub.2017.09.041Ulrich, L. E., Koonin, E. V., & Zhulin, I. B. (2005). One-component systems dominate signal transduction in prokaryotes. Trends in Microbiology, 13(2), 52–56. https://doi.org/10.1016/j.tim.2004.12.006Wall, M. E., Hlavacek, W. S., & Savageau, M. A. (2004). Design of gene circuits: Lessons from bacteria. Nature Reviews Genetics, 5(1), 34–42. https://doi.org/10.1038/nrg1244Yawata, Y., Carrara, F., Menolascina, F., & Stocker, R. (2020). Constrained optimal foraging by marine bacterioplankton on particulate organic matter. Proceedings of the National Academy of Sciences, 117(41), 25571–25579. https://doi.org/10.1073/pnas.2012443117Yawata, Y., Cordero, O. X., Menolascina, F., Hehemann, J.-H., Polz, M. F., & Stocker, R. (2014). Competition–dispersal tradeoff ecologically differentiates recently speciated marine bacterioplankton populations. Proceedings of the National Academy of Sciences, 111(15), 5622–5627. https://doi.org/10.1073/pnas.1318943111Zöttl, A., & Yeomans, J. M. (2019). Enhanced bacterial swimming speeds in macromolecular polymer solutions. Nature Physics, 15(6), 554–558. https://doi.org/10.1038/s41567-019-0454-3